# A blended modeling framework for real-time design and verification of safety-critical embedded systems

Misbah Mehboob Awan*, Muhammad Waseem Anwar,
Wasi Haider Butt, Farooque Azam

Department of Computer and Software Engineering, College of Electrical and Mechanical Engineering, National University of Sciences and Technology (NUST), Islamabad, Pakistan

These authors contributed equally to this work.
* mmehboob.cse19ceme@ce.ceme.edu.pk

## Abstract

Embedded systems often require multiple representations for design, verification, and implementation, ranging from low-level programming languages to high-level formal models and domain-specific abstractions. Generally, synchronization among different representations or notations is achieved manually, a process that is labor-intensive and prone to mistakes, adversely impacting productivity and time-to-market objectives. Despite existing tool support, there remains a lack of unified, automated mechanisms that ensure semantic consistency across heterogeneous modeling and programming notations. This article presents a scalable blended modeling framework that automates the synchronizations across an extensible set of notations using bidirectional transformations. This facilitates the system development, comprising design and verification aspects of safety-critical embedded systems, using various notations interchangeably. The applicability of the proposed framework is demonstrated using four distinct representations: C, SystemVerilog, Timed Automata, and a domain-specific modeling language. The framework supports a notation-agnostic design flow, allowing development to begin from any of the supported languages. This enables seamless transitions across notations based on design or verification needs. Validated through two industrial case studies, a ventilator system and a cruise control system, the framework achieved high round-trip transformation accuracy with minimal information losses in edge cases such as language-specific keywords. Performance evaluations revealed low transformation latency and modest memory consumption, supported by efficient Abstract Syntax Tree (AST) traversal. This research lays the groundwork for the standardization of model-to-code, code-to-model, and code-to-code transformations, significantly reducing manual engineering effort and improving the reliability and agility of embedded systems design and verification processes.

**Data availability statement:** All MRED Project files are available from the Github repository (https://github.com/ MisbahAwan/MRED_Project/tree/main).

**Funding:** This work is partially supported by the Higher Education Commission, Pakistan, through the NRPU MRED project under Grant No. [20-15651]. The funders had no role in study design, data collection and analysis, decision to publish, or preparation of the manuscript. There was no additional external funding received for this study.

**Competing interests:** The authors have declared that no competing interests exist.

## 1 Introduction

Embedded systems have been a growing trend over the past decades in key industries such as consumer electronics, aerospace, industrial automation, and automotive systems [1]. However, their design and verification present numerous challenges. **Design challenges** arise from real-time requirements, integration of heterogeneous components, resource constraints, and the need for cross-domain expertise. On the other hand, **verification challenges** include managing temporal constraints, fragmented tools, disjoint representations, resource constraints, budget limitations, and the complexity of formal methods.

Embedded systems have highly specific and diverse requirements, such as low power consumption, high performance, real-time responsiveness, and cost efficiency. Meeting these requirements simultaneously can be difficult. However, designing embedded systems requires expertise across multiple domains. Using different tools for design (e.g., Unified Modeling Language (UML), SysML (Systems Modeling Language), SystemVerilog [2]) and verification (e.g., Timed Automata [3]) often leads to inconsistencies and redundant efforts. They require specialized knowledge to transform design representation into verification representation. Formal techniques like model checking and theorem proving offer high assurance but are often resource-intensive and time-consuming. These challenges, coupled with limited budgets and development constraints, can hamper productivity and delay project delivery.

Embedded system development draws upon diverse technologies and notations to address different facets of design and verification. Abstract modeling tools like UML and meta-modeling frameworks support early-stage design by improving clarity and reuse. Meanwhile, languages like C and SystemVerilog are indispensable for actual implementation, especially on constrained hardware platforms. SystemVerilog also plays a crucial role in verification through features like assertions and Universal Verification Methodology (UVM). Timed automata contribute by modeling temporal behaviour for real-time validation. However, the use of these tools in isolation can create bottlenecks, inconsistencies, and longer verification cycles. Integrating these approaches within a unified framework can improve design-verification alignment, reduce manual overhead, and better meet the rapid timelines required in modern product development.

Although Domain Specific Modeling Languages (DSMLs) simplify system modeling at an abstract level, the transformation into executable representations like C is often one-way, limiting the ability to refine or verify changes iteratively. Reverse transformation to the abstract model becomes challenging, complicating system updates and traceability. Additionally, performing dynamic verification requires transformation into SystemVerilog, demanding further effort, time, and resources. Yet another transformation into specialized formal languages, such as Timed Automata, is needed for formal verification, often requiring a separate design effort. These fragmented and manual transitions across multiple representations create inefficiencies and inconsistencies in the development process. To address these challenges, this research identifies the need for a real-time, unified framework that ensures bidirectional synchronization among diverse notations. The proposed framework, though demonstrated

with synchronization among four representations including DSML, C, SystemVerilog, and Timed Automata, is generic and scalable, capable of supporting synchronization across *n* number of notations for embedded system design and verification.

Innovative solutions such as **model-driven frameworks**, integrated workflows, and automated tools effectively address the challenges of embedded system design and verification by enhancing consistency, reducing redundancy, and streamlining processes. These approaches allow designers to focus on functionality while ensuring robust and reliable system verification. Consequently, **blended modeling** and model-driven frameworks have emerged as transformative methodologies that simplify embedded system development through structured techniques for managing complexity, maintaining consistency, and automating verification. Blended modeling [4] overcomes the fragmented nature of traditional workflows by integrating multiple paradigms, languages, and tools into a unified framework. By combining abstract representations like UML with concrete implementation and verification models such as SystemVerilog and Timed Automata, it delivers a comprehensive system perspective that fosters shared understanding and smoother collaboration across hardware, software, and verification domains. Moreover, blended modeling ensures coherence across heterogeneous notations by automatically synchronizing changes between related views (e.g., C, DSML, Timed Automata), minimizing manual rework and transformation errors, and enhancing the reliability of the development process. Through the integration of both visual and textual representations, it bridges communication gaps among multidisciplinary teams and supports empirical coverage of selected subsets of models, thereby advancing the efficiency and effectiveness of the overall embedded system lifecycle.

This paper presents a blended modeling framework that bridges domain-specific abstractions with executable and verifiable models. The framework maintains real-time bi-directional consistency across notations, enabling dynamic switching between modeling views without redundancy or semantic drift. Its scalable architecture allows for the integration of additional notations, ensuring adaptability to diverse embedded system contexts and accelerating the design–verification cycle through unified, synchronized transformations.

The framework establishes traceability from requirements to implementation and verification, supporting iterative feedback loops that enable continuous refinement of designs based on validation outcomes. By automating transitions and streamlining modeling workflows, the framework significantly reduces the time and effort required for embedded system development, aligning well with the industry's growing need for agile and efficient solutions.

The major contributions of this paper are as follows:

- The development of a blended modeling framework (Sect 3) that integrates the design and verification of embedded systems by enabling synchronization among multiple notations. This includes specifying language subsets for each representation, ensuring they capture the essential syntax and semantics needed for reliable transformation and validation.
- The implementation of a transformation engine that applies defined high-order transformation rules to the specified language subsets, enabling seamless bidirectional transformations across multiple representations (Sect 3). Although this paper focuses on four notations, the framework is inherently scalable, designed to support an arbitrary number of representations. A graphical user interface is provided (Sect 4) to facilitate real-time visualization and transformation.
- The framework is validated (Sect 5) through two industrial case studies: a ventilator system and a cruise control system. These applications demonstrate the practicality and effectiveness of the approach in addressing real-world challenges, resulting in substantial productivity gains. Comparative evaluations indicate that the framework substantially reduces development effort by automating transformations and minimizing manual tasks.

These contributions address fundamental challenges in embedded system design and verification by providing an extensible, automated, and synchronized approach. By precisely defining language subsets for each notation (both abstract and concrete) and implementing a robust transformation engine, the framework supports consistent and interoperable development across all supported representations. Its interactive graphical interface further simplifies the user experience, enabling real-time, low-overhead transformations. Through validation on real-world systems, the framework

proves its capability to handle complex embedded systems while maintaining flexibility for future expansion to additional modeling domains.

Fig 1 illustrates the end-to-end workflow of the proposed blended modeling framework for embedded systems design and verification. It begins with multiple input notations, ranging from abstract syntax models like DSML to three concrete syntactic representations and potentially any $n^{th}$ notation. Each notation undergoes concept or subset identification to isolate key elements relevant for transformation. These are unified through grammar definitions, forming the basis for parsing and transformation. High-order transformation rules are then applied to enable consistent, bidirectional mapping between representations. A Graphical User Interface (GUI) editor facilitates real-time interactions, allowing designers to visualize, edit, and switch seamlessly between synchronized views. The synchronized outputs, maintained across all notations, can then be directly fed into industry-standard verification and validation tools like UPPAAL, QuestaSim, and C compilers. This layered, modular structure highlights the framework's generic, scalable, and automation-friendly design, making it adaptable to any number of modeling languages while supporting agile and consistent embedded systems development.

## 2 Literature review

The design and verification of embedded systems continue to pose significant challenges due to the complexity, safety-critical requirements, and heterogeneous nature of such systems [5]. Numerous studies have addressed various facets of these challenges, ranging from design abstraction to runtime validation [5–8]. Among the programming languages used, C remains dominant in embedded system development, particularly due to its efficiency and hardware-level control [9]. As noted in [10] , C continues to be the leading language for the development of IoT and embedded applications, underscoring its foundational role in embedded system design.

For dynamic verification, languages such as Verilog and SystemVerilog have gained widespread adoption, especially in the automotive and hardware design domains [7,8]. A detailed state-of-the-art review on UART design and verification presented in [6] highlights the critical role SystemVerilog plays in the dynamic verification of embedded systems, particularly in enabling assertion-based and simulation-driven testing.

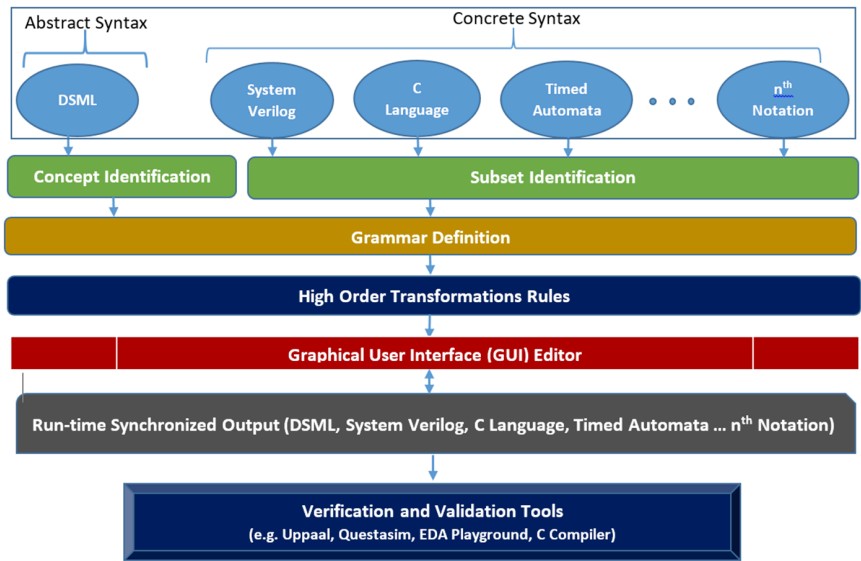

**Fig 1**. Blended modeling workflow for automated bidirectional transformations in embedded systems design and verification.

In parallel, significant progress has been made in the application of formal methods for ensuring the correctness of embedded systems. Formal verification techniques offer mathematical rigor by allowing the exhaustive verification of system properties through model checking [11]. For instance, [12] demonstrates how formal modeling and verification enhance the reliability of distributed systems through structured validation. Tools like UPPAAL support real-time model checking by enabling the modeling of timed automata and verifying temporal logic properties [13,14]. A recent study [15] provides a comprehensive overview of UPPAAL-assisted formal verification methods and illustrates their applicability across diverse domains, including the dynamic verification of embedded systems. These developments collectively emphasize the need for integrating design, dynamic, and formal verification approaches within a unified framework.

Each representation, including C for low-level implementation, SystemVerilog for simulation and verification, and UPPAAL for formal analysis, contributes uniquely to the design and validation of real-time embedded systems. C enables efficient, hardware-near execution, while SystemVerilog supports dynamic verification and early bug detection through simulation and assertions. UPPAAL complements this with formal model checking to validate time-critical and safety-related properties with mathematical rigour. Collectively, these notations ensure correctness, performance, and safety, especially in high-stakes domains. However, the absence of coordination across them often leads to duplicated effort, semantic mismatches, and longer development cycles. This underscores the necessity for an integrated framework that synchronizes these representations throughout the system lifecycle.

Model-Driven Development (MDD) has become a cornerstone for managing the complexities of embedded systems design, enabling abstraction through high-level models. Schmidt et al. [16] demonstrated MDD's ability to reduce complexity via automated code generation, while Mellor's [17] Model-Driven Architecture (MDA) standardized platform-independent modeling. **Anwar et al.** [18] extended MDD for verification by introducing SVOCL, an OCL extension for SystemVerilog, enabling automated consistency checks between models and code. However, their work focuses on single-language transformations and lacks support for **multi-notation bi-directional transformations**.

Blended modeling [4] has emerged as a promising approach to address the fragmented nature of embedded systems development. By enabling the integration of multiple representations in a unified framework, blended modeling facilitates seamless transitions between design and verification stages. MDE complements this approach by emphasizing the use of abstract models (abstract syntax) as primary artifacts in the development process. These models are iteratively refined and transformed into various representations (concrete syntaxes), such as hardware description languages or formal methods, through automated transformations. Further, blended modeling has become a key approach for embedded systems design and verification, with few studies [18,19] proposing frameworks to integrate heterogeneous representations. However, these existing works do not cater for the embedded systems design and verification notations. In this context, selecting relevant state-of-the-art blended modeling works is critical to contextualize the research gap.

Several frameworks and tools have been proposed to address the challenges of embedded systems design and verification. It is to be noted that our study focuses specifically on the automatic generation of **horizontal** model transformations. It does not examine approaches for the automatic generation of vertical model transformations. A recent SLR [20] on blended modeling tools and frameworks has been conducted. They have identified 26 tools. Most of them use multiple concrete notations for a single underlying abstract syntax. However, they either lack effective inconsistency tolerance mechanisms or do not utilize blended modeling features to improve user experience via bi-directional transformations.

A lot of work has been done in blending the modeling of textual and graphical notations. They provide a limited set of features as only one notation is editable and the other is read-only [21–23]. Hence, they don't allow editing the model via multiple notations. Few language-specific solutions have been proposed. Maro et al. [24] propose a solution for integrating graphical and textual editors for a UML profile-based domain-specific language (DSL). Their work focuses on generating a textual editor from an existing graphical editor and enabling seamless switching between the two views. To achieve this, the UML profile-based DSL is first transformed into an Ecore model using ATL transformations. The Ecore model is then utilized by the Xtext plugin to generate the textual editor. Synchronization between the graphical and textual views is facilitated through ATL transformations, ensuring consistency across representations. However, their approach is **limited**

**to UML profiles** and does not support low-level languages like C or SystemVerilog, nor does it integrate formal verification tools like UPPAAL. In contrast, our framework employs an **abstract syntax**, enabling bidirectional transformations between multiple **concrete syntaxes**, thus bridging the gap between high-level design and low-level implementation for the design and verification of embedded systems.

Addazi and Ciccozzi [25] present a proof-of-concept implementation for blended modeling of UML and UML profiles, combining graphical and textual notations. Their solution leverages the Eclipse Modeling Framework (EMF), Xtext, and Papyrus, with a single underlying abstract syntax and two notations (graphical and textual) sharing a common UML resource. Synchronization is achieved through serialization and deserialization operations between Xtext and UML models. While their approach improves user performance compared to single-notation modeling, it remains **restricted to UML-based DSLs** and lacks support for formal verification or runtime adaptability. Our framework, on the other hand, integrates **formal verification** and provides an **interactive GUI** for real-time transformations, making it more versatile and user-friendly.

Lazar [26] integrates the Alf textual editor with the Eclipse UML tree-based editor to create fUML models. However, synchronization between the textual and graphical representations is on demand. This approach limits real-time consistency and requires explicit user intervention to propagate updates. In contrast, our framework ensures **real-time bidirectional synchronization** through an ANTLR-driven transformation engine, ensuring consistency across all representations.

Scheidgen [27] introduces embedded textual editors as an add-on feature for graphical editors. When a user selects a model element for editing, the embedded textual editor generates an initial representation, which the user can modify. Parsing operations are then used to create updated model elements. However, synchronization is also **on-demand**, as changes to the underlying model are only applied when the user commits them and closes the textual editor, delaying real-time updates. Our framework addresses this limitation by enabling **runtime transformations** through an interactive GUI, allowing users to switch between multiple representations with minimal delay and ensuring immediate consistency.

Latifaj et al. [19] introduced higher-order transformations (HOTs) to generate synchronization infrastructures for blended models (e.g., UML to timed automata). This work **automates synchronization mechanisms across multiple notations,** regardless of whether they share the same abstract syntax or belong to different languages. Their solution is designed for modeling environments based on the Eclipse Modeling Framework (EMF) and DSMLs defined using EMF's meta-metamodel, Ecore. Hence, it is specific to UML and DSML unidirectional transformations.

Ciccozzi et al. [28]emphasized runtime representation switching to accelerate design-verification cycles. However, existing frameworks (e.g., **Anwar et al. [29]**, Latifaj et al.'s HOTs [19]) focus on high-level formalisms and lack support for bidirectional and real-time transformations between low-level languages (C, SystemVerilog) and verification tools like Uppaal. Recent work on EAST-ADL blended modeling has demonstrated the feasibility of real-time synchronization across heterogeneous views. **Anwar et al. [30]** integrated Xtext and EATOP to achieve runtime bidirectional synchronization between textual and graphical notations using EAXML as the common format. Validated through a Volvo car wiper case study, the framework effectively maintained timing and variability consistency, showcasing the potential of blended modeling for industrial applications. However, its applicability was **limited to two notations** and **lacked dedicated tool support**, restricting its scalability to broader language domains such as C, SystemVerilog, and Timed Automata. This highlights the need for a more extensible**,** tool**-**supported blended framework capable of ensuring real-time, semantically consistent synchronization across multiple design and verification notations. Our proposed research bridges this gap by enabling runtime transformations among abstract and concrete notations, ensuring traceability and reducing manual rework. In our proposed solution, we are following a parser-based approach, which ensures the correctness of the syntactic aspects of transformations. The other approach is the direct AST updation approach [31] , which compromises consistency and correctness among models. EAST-ADL/AutoSAR [32] provides a standardized development environment for automotive systems, while UPPAAL supports formal verification of timed automata models. However, these solutions are often domain-specific, lack support for multiple representations, or do not provide runtime transformation capabilities.

## 2.1 Research gap

Despite notable progress in model-driven, blended, and formal methods for embedded system design and verification, several critical limitations remain unaddressed. First, most existing frameworks lack a **centralized abstract syntax** to maintain semantic consistency across heterogeneous notations. Approaches such as EAST-ADL/AutoSAR [32] and Anwar et al. [18,29] remain **domain-specific** or **language-bound**, restricting interoperability across modeling and verification environments. Second, frameworks like those of Latifaj et al. [19] and Maro et al. [24] emphasize **high-level abstractions** (e.g., UML-based DSMLs) while neglecting **low-level implementation languages** such as C and SystemVerilog, which are indispensable for embedded systems development. Third, although bidirectional transformation has been acknowledged as essential for maintaining synchronization between design and verification models, **true round-trip consistency** remains largely unaddressed. Solutions based on AST-level synchronization (e.g., Atkinson et al. [31]) are prone to semantic drift, whereas others (e.g., Addazi & Ciccozzi [25]) support only partial or delayed synchronization. Recent efforts in **EAST-ADL blended modeling** [30] demonstrate real-time synchronization capabilities, it remains limited to two notations and lacks a standalone transformation tool or extensibility toward additional representations. Moreover, its reliance on a domain-specific EAXML meta-model restricts scalability and generalization to other embedded domains.

Moreover, most frameworks do not offer **real-time or runtime transformations** and lack **interactive graphical interfaces**, limiting their industrial scalability and adoption. Approaches like those of Scheidgen [27] and Lazar [26] depend on on-demand synchronization, delaying consistency propagation, and reducing efficiency. A further gap exists in integrating **formal verification tools** such as UPPAAL within blended environments, as formal verification is often treated as a post-design activity rather than a continuous design component. Finally, the **validation scope** in current literature remains narrow, with most works limited to proof-of-concept or academic examples rather than comprehensive **empirical coverage of selected subsets** validated through industrial case studies.

To overcome these limitations, the present research introduces a **parser-based blended modeling framework** that unifies C, SystemVerilog, UPPAAL, and DSML under a shared abstract syntax, enabling **runtime bidirectional synchronization** and **round-trip transformations**. This approach ensures syntactic and semantic consistency, strengthens traceability between notations, and achieves practical scalability validated through real-world embedded system case studies.

## 3 Proposed framework

The proposed framework introduces a blended modeling approach that facilitates real-time transformations for the design and verification of embedded systems. This comprehensive methodology is grounded in the principles of model-driven engineering (MDE) and is utilized to support bidirectional transformations. By managing multiple system representations, the framework ensures consistency, scalability, and efficiency in embedded systems development.

The framework, presented in Fig 2, introduces a unified environment for designing and verifying complex embedded systems through *simultaneous support* of multiple modeling and programming languages. It is designed to be generic and extensible, supporting a wide range of notations for system development. For proof of concept, we demonstrate the framework using four notations: **Domain-Specific Modeling Languages (DSMLs)**, **C**, **SystemVerilog**, and **Timed Automata**, enabling designers to represent system behavior and structure in diverse paradigms (for example, software-centric, hardware-aware, or formal-method-driven). Crucially, the framework allows **bidirectional runtime switching** between representations. For instance, a DSML model can dynamically transform into SystemVerilog code or Timed Automata states, and vice versa. By eliminating manual transformation bottlenecks, the proposed framework significantly accelerates design iteration and verification cycles.

### 3.1 Architectural components of proposed framework

The proposed **Blended Modeling Framework for Design and Verification of Safety Critical Embedded Systems** (Fig 2) integrates multiple modeling paradigms into a cohesive structure that supports **real-time, bi-directional, and multi-way transformations**. This architecture is divided into three major components:

**Language subset evaluation and selection.** This section manages the **abstract and concrete syntaxes** of the notations. The abstract syntax is unified under a **Domain Specific Modeling Language (DSML)**, which captures the core semantics of the system regardless of its representation. The concrete syntaxes correspond to multiple notations such as **C Language, SystemVerilog, Timed Automata**, and potentially many others. Each concrete syntax conforms to the overarching DSML, ensuring a consistent semantic foundation across views. This employs static analysis, dependency tracing, and semantic profiling to identify minimal yet sufficient subsets. The result is a **streamlined subset** with unified syntax and semantics, enabling consistent and reliable transformation across different languages.

**Transformation engine.** This component manages the **bi-directional transformation rules** for each notation. The transformation engine works by applying these rules over **notation subsets and grammar**, which include **lexical and syntactic definitions** of each concrete notation. The engine supports seamless switching between representations while maintaining semantic fidelity. The **Mapping Rules** interface links the transformation engine with the editor, enabling consistent synchronization across notations. The engine encodes *one-to-one*, *one-to-many*, and *many-to-many* mapping rules directly into transformation logic, ensuring syntactic and semantic fidelity during conversions. For example, a SystemVerilog function block may map to a Timed Automata template via AST traversing.

**Multi-representation editor.** Integrates a rich editor to visualize and manipulate designs in any supported language. The editor performs **model-to-text** (DSML→SystemVerilog) and **text-to-model** (C→Timed Automata) transformations in real time. Anyone can interact with the system via various notation-specific views, such as C View, SystemVerilog View, etc. These views are synchronized in real time using bi-directional transformation rules. This ensures that a change in one view automatically reflects across others. The High-Order Transformations block at the bottom handles multi-way bi-directional transformations, allowing for more complex and scalable interactions across multiple notations simultaneously.

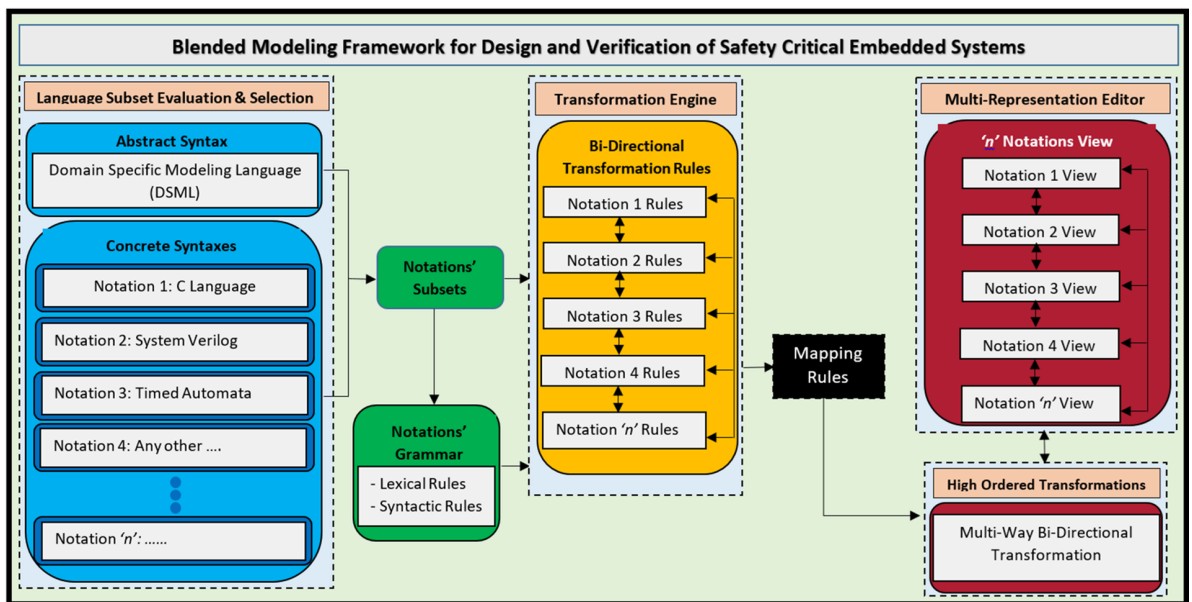

**Fig 2**. **Architecture of the blended modeling framework for design and verification of real-time embedded systems.**

Within the proposed architecture, data exchange between heterogeneous notations occurs through a well-defined intermediate representation inside the **Transformation Engine** and **Multi-representation Editor**. The exchange process is not based on direct file-level conversion but on the shared DSML meta-model, which serves as a neutral data bridge between C, SystemVerilog, Timed Automata (UPPAAL), and DSML structures. When a user modifies a model in any notation, the Transformation Engine serializes the change into an intermediate AST form conforming to the meta-model and propagates it across other notations through transformation rules Sect 3.4. The **Language Subset Evaluation and Selection** layer defines the syntactic and semantic scope for each notation, but does not perform data transfer itself. The **Multi-representation Editor** then visualizes and synchronizes these updates in real time, allowing consistent co-editing and validation. This **model-centric data exchange process** ensures semantic integrity, supports bidirectional synchronization, and enables round-trip engineering across all supported representations.

The proposed blended modeling framework integrates an abstract meta-model (DSML) with multiple concrete syntaxes to address embedded system design and verification challenges. The meta-model provides a unified semantic base, while the concrete syntaxes offer domain-specific views for designers and verification engineers. Through High-Order Transformations (HOTs), the framework automates bidirectional translations between models and code, ensuring correctness-by-construction and minimizing manual validation. This scalable and synchronized environment unifies formal, software, and hardware paradigms, streamlining the design, verification, and evolution of complex embedded systems.

## 3.2 Abstract syntax meta-model (Domain Specific Modeling Language DSML)

To formalize the abstract syntax, we developed a foundational *meta-model (Fig 3)* that encapsulates the structural, behavioral, and temporal semantics common across all concrete syntaxes (e.g., C, SystemVerilog, Timed Automata, etc.). This meta-model acts as a *universal semantic backbone*, explicitly defining core entities such as system, components, variables, and statements, along with their interrelationships and operational rules. By abstracting domain-agnostic constructs, the meta-model ensures semantic consistency while accommodating domain-specific extensions required for each concrete syntax. This approach enables seamless mapping between heterogeneous syntaxes, preserves cross-domain interoperability, and provides a unified framework for verification tools to operate upon. Domain-specific details are integrated as specialized annotations within the meta-model, ensuring fidelity to their native semantics while maintaining alignment with the abstract layer. The result is a robust, extensible foundation that reduces ambiguity in cross-tool workflows and supports automated transformations via HOTs.

The meta-model serves as a unified semantic foundation for representing programs in **C**, **SystemVerilog**, and **Timed Automata**. It abstracts domain-agnostic constructs while accommodating language-specific features through hierarchical composition. Fig 3 illustrates its structure, which is organized as follows:

**3.2.1 Constructs. System class.** The System class serves as the root entity, encapsulating the global structure of a program. It aggregates:

- **Component**: Represents modular units such as functions (C), modules (SystemVerilog), or automata (Timed Automata).
- **Comment**: Captures annotations for documentation and readability.
- **Preprocessor/Include Directives**: Manages cross-file dependencies, macros, and library inclusions.
- **Declaration**: Defines variables, signals, or functions.
  *The declaration* further refines into:
  - **Attribute**: Metadata.
  - **Enum**: Enumerated types for state or signal definitions.
  - **Function Prototype**: Specifies function signatures, including parameters and return types.

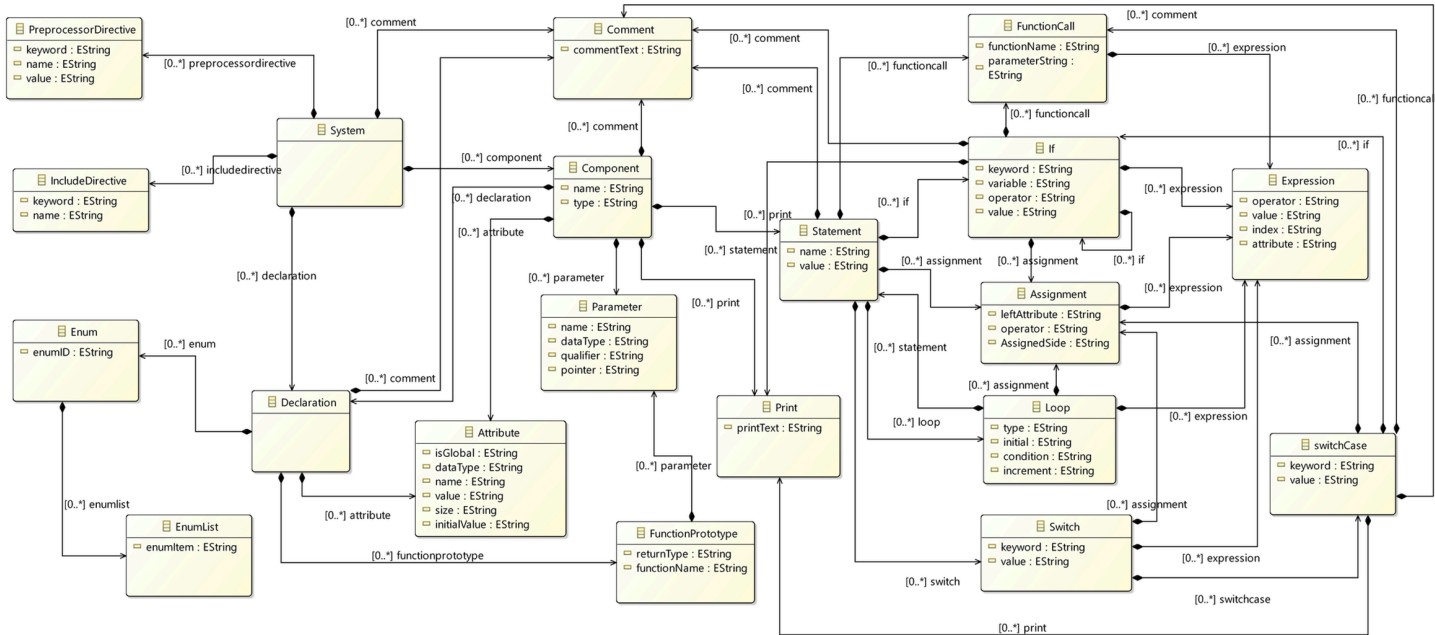

**Fig 3. Abstract syntax metamodel for blended modeling in embedded systems.**

**Component class.** The Component class represents reusable, self-contained units of functionality and encapsulates the functional flow of a program, aggregating:

- **Declaration**: Local variables, Global variables, signals, or registers.
- **Parameter**: Configurable constants (e.g., #define in C, parameter in SystemVerilog).
- **Print**: Output operations for debugging or logging (e.g., printf, $display).
- **Statement**: Defines the control and data flow within the components and their Execution logic. (Detailed below).

   **Statement class.** The Statement class forms the backbone of program logic, capturing execution flows across languages. The Statement hierarchy defines control and data flow. It aggregates:

- **FunctionCall**: Invokes procedures or methods.
- **If-Else**: Conditional branching, using **Expression** to evaluate logic (e.g., if (x != 0) {...}).
- **Assignment**: Variable updates, driven by **Expression** (e.g., x = y + 5;).
- **Loop**: Iteration constructs (for, do, while), which recursively nest **Statement** blocks.
- **Switch**: Multi-way branching, aggregating **SwitchCase** instances.
- Each **SwitchCase** may contain nested **Statement** blocks.

   **Cross-language consistency.**

- **Expression**: A shared construct for arithmetic, Boolean, or temporal logic (e.g., C expressions, SystemVerilog assertions, Timed Automata guards).
- **Recursive Structure**: Loops and conditional blocks reuse the Statement class, enabling nested logic across languages.

The meta-model, developed in the Eclipse Modeling Framework (EMF) (Fig 3), provides a common semantic backbone that bridges syntactic differences among C, SystemVerilog, and Timed Automata. It defines shared abstractions such as Declaration, Statement, and Expression to enable bidirectional mappings and maintain consistency during runtime switching and automated transformations. Serving as a centralized abstract syntax, it encapsulates the semantic essence of all supported languages through components like variables, control statements, and declarations, ensuring correctness and integrity across transformations. The hierarchical structure of the meta-model places System at the root, branching into Component, Declaration, and Statement, which further expands into FunctionCall, If-Else, Assignment, Loop, and Switch, with explicit aggregation and recursive relationships as shown in Fig 3.

### 3.3 Concrete Syntaxes (C Language, SystemVerilog, Time Automata)

In our blended modeling framework, concrete syntaxes represent the tangible, language-specific formulations of system models prior to their abstraction and unification. Although the framework is inherently extensible and capable of accommodating any number of concrete syntaxes, we focus on three prominent representations: **C**, **SystemVerilog**, and **Timed Automata**. These have been deliberately selected due to their significant roles in the design and verification of embedded systems. C facilitates low-level imperative programming with direct access to memory and hardware, SystemVerilog offers a robust environment for hardware modeling and simulation, and Timed Automata provide formal semantics for modeling and verifying time-critical behaviors. Each of these languages is defined using context-free grammars and is systematically mapped to a unified abstract syntax meta-model, enabling automated, traceable, and consistent transformations across different modeling levels.

**3.3.1 Definition of language subsets.** The full syntactic and semantic complexity of languages like C, SystemVerilog, and Timed Automata often introduces challenges when integrating them into a unified transformation framework. To overcome these challenges, we adopt a subset selection process that distills each language to its most essential constructs. This process is driven by three primary criteria: relevance, simplicity, and consistency. First, we retain only those constructs that are critical for accurate modeling, design verification, and ensuring semantic fidelity across representations. Second, by eliminating rarely used or overly complex features, we simplify the grammars and the mapping process to the abstract meta-model, thereby reducing ambiguity and easing maintenance. Ensuring consistency across language subsets is essential for effective bidirectional transformations. Shared concepts such as control flow, expressions, and data types must be uniformly represented. The subsets are selected through static analysis, dependency tracking, and semantic profiling to keep them both minimal and sufficiently expressive.

The methodology begins by defining concise subsets for each language relevant to embedded systems:

- **C Subset**: Captures fundamental constructs like data types, control structures, functions, print statements, assignment statements, etc., streamlining the transformation without compromising essential functionality.
- **SystemVerilog Subset**: Includes elements needed for system behaviour and verification, such as modules, data types, variable assignments, control flows, etc.
- **Timed Automata Subset**: Aligns with UPPAAL's structure, focusing on transitions, states, clocks, and invariants to support formal verification of real-time behaviour.

These tailored subsets simplify parsing and transformation while ensuring fidelity and precision. Tables 1, 2, 3 present the selected constructs used in our framework.

**C Language subset selection.** Table 1 outlines the selected C language constructs used in our transformation framework. These elements are carefully chosen to balance modeling precision with computational expressiveness, ensuring compatibility with formal verification needs.

**Table 1. C Language subset.**

| Category | Construct |
|---|---|
| Data Types | int, char, float, double |
| | long, short |
| | signed, unsigned |
| | struct, union |
| | enum |
| Control Structures | if-else |
| | switch-case |
| | for, while, do-while loops |
| | break, continue |
| | goto |
| Functions | Function definition |
| | Function declaration |
| | Function call (with/without parameters) |
| | Recursion |
| Pointers | Pointer variables |
| Arrays | One-dimensional arrays |
| Preprocessor | #define, #include |
| | #ifdef, #ifndef, #endif |
| Variables | Extern variables |
| Assignments | Assignment with constant |
| | Assignment with variable |
| | Assignment with function call |

**Table 2. SystemVerilog subset.**

| Category | Construct |
|---|---|
| Data Types | bit, logic, reg |
| | integer, real |
| | enum, typedef |
| Control Structures | if-else |
| | case |
| | loop |
| | break, continue |
| Procedural Blocks | initial, always |
| | assign, deassign |
| Assignments | variable assignment |
| | assignment through function |
| | constant assignment |
| Function call | with parameters |
| | without parameters |

The selected subset focuses on essential constructs for transformation and verification, intentionally excluding concurrency and low-level operations to maintain alignment with formal methods and manageable complexity. The current prototype supports both simple and moderately complex case studies, reflecting practical relevance through alignment with industry needs. While streamlined for feasibility, the framework remains extendable to incorporate additional constructs in future enhancements.

**SystemVerilog language subset selection.** Table 2 outlines the selected SystemVerilog constructs used for modeling, simulation, and verification. The subset is chosen to maintain a balance between descriptive capability and practical transformation, supporting compatibility with formal verification processes.

**Table 3.** **Timed automata subset.**

| Category | Construct |
|---|---|
| Locations | States (committed, urgent) |
| Transitions | Synchronization (channels) |
| | Guards |
| | Updates |
| | Invariants |
| Data Types | int, bool |
| | clocks |
| | arrays |
| Control Structures | if-else |
| | loops (while, for) |
| | functions |

**Timed Automata (UPPAAL) subset selection.** The UPPAAL subset selection (Table 3) focuses on timed automata modeling constructs that facilitate accurate system behaviour representation. By focusing on these constructs, the subset ensures a structured transformation process while maintaining alignment with UPPAAL's modeling paradigms.

**3.3.2 Empirical coverage of selected subsets.** While the subset definitions in Tables 1, 2, 3 were guided by relevance, simplicity, and consistency, it is essential to empirically validate their adequacy against representative industrial codebases. To this end, we analyzed established benchmark suites and industrial case studies commonly employed in embedded system research and practice. Table 4 summarizes the observed coverage of our subsets relative to the constructs encountered in these sources.

To substantiate the practical adequacy of the defined subsets, an empirical analysis was conducted using representative programs from two public benchmark suites, MiBench [33] and WCET [34], and two industrial case studies: Automotive Engine Control and Medical Ventilator Controller. These selections ensured both research-standard validation and industrial relevance. The objective was to evaluate how effectively the subsets capture frequently occurring syntactic constructs in real-world embedded code while excluding rarely used or verification-incompatible features.

**Analysis Procedure:** Each benchmark was parsed using custom front-end parsers built with ANTLR grammars for C and SystemVerilog. The UPPAAL subset was formulated to remain compatible with C and SystemVerilog constructs, ensuring consistency across design and verification levels. The DSML subset provides a generic abstraction layer, encapsulating common semantics across all notations. This structure ensures that transformations among C, SystemVerilog, UPPAAL, and DSML remain semantically aligned and practically verifiable through benchmark-based evidence. The process involved:

• Enumerating all syntactic constructs (loops, conditionals, declarations, expressions, etc.) in each benchmark.
• Mapping these constructs to corresponding elements in the defined subsets (Tables 1–3).
• Identifying unsupported or excluded constructs (e.g., dynamic memory, recursion, hardware-specific primitives).

The ratio of supported constructs to total constructs yielded an *estimated coverage percentage*, reflecting the representational adequacy of each subset.

Coverage values were estimated by statically analyzing the selected programs using the subset-aware parser. Each construct type was counted based on its syntactic occurrence and verified for compatibility with the defined subsets. Across MiBench (ADPCM, Dijkstra, CRC32) and WCET (Automotive, FFT), the subsets covered approximately 70–85% of observed constructs, excluding low-level or hardware-specific features deliberately omitted to ensure alignment with formal verification semantics. Both industrial case studies achieved full coverage as their models were designed within subset boundaries.

**Table 4**. Empirical coverage of selected language subsets across representative benchmarks.

| Benchmark/Case Study | Representative Programs/Modules | Estimated Coverage Range (%) | Key Excluded Features |
|---|---|---|---|
| MiBench Embedded Suite [33] | ADPCM, Dijkstra, CRC32 | 70–80% | Dynamic memory, inline assembly, thread-level concurrency |
| WCET Benchmark Suite [34] | Automotive, FFT | 75–85% | Recursive calls, low-level timing primitives, hardware-specific optimizations |
| Automotive Engine Control Case Study | Start Ventilator Mode, Mode Shift Control Module | 100% | None (subset-compliant) |
| Medical Ventilator Controller Prototype | Speed Monitoring Module | 100% | None (subset-compliant) |

This empirical evaluation confirms that the proposed subsets are semantically coherent and practically expressive for embedded system modeling and transformation tasks. By combining benchmark-driven analysis with domain-specific validation, the study provides both quantitative and qualitative evidence supporting the completeness and generalizability of the subset definitions. The excluded constructs primarily correspond to advanced concurrency features (threads, tasks, interrupts) and certain low-level optimizations that were deliberately omitted to maintain tractability and alignment with formal verification tools. These findings empirically validate the suitability of our subset definitions for a wide range of embedded system applications, while also highlighting clear directions for future extensions.

**3.3.3 Grammar definition.** Defining a formal grammar is essential for parsing and transforming programming and modeling languages, as it guarantees syntactic correctness and enables feasible transformations. In this work, we designed a concrete syntax grammar for C, SystemVerilog, Timed Automata (UPPAAL), and DSML, drawing on context-free grammars (CFGs), attribute grammars, and parsing expression grammars (PEGs). ANTLR (Another Tool for Language Recognition) was adopted due to its LL(*) parsing strategy, modular design, error handling, and automated parse tree generation, making it suitable for unifying the syntactic treatment of software, hardware, and formal models within a single framework. This ANTLR-based grammar forms the backbone of our transformation engine by ensuring precise syntax definitions, scalability, and seamless cross-domain integration. To maintain readability, complete grammar specifications and transformation rules are not reproduced in full; instead, the C grammar, its rules, and illustrative AST examples are provided in Appendix 9, while the SystemVerilog, UPPAAL, and DSML grammars remain available in our GitHub repository [35]. This division preserves clarity in the manuscript while ensuring transparency and reproducibility.

## 3.4 Bi-directional transformation rules

The core of the proposed framework lies in its ability to perform real-time bi-directional transformations between the selected representations. Although the proposed framework supports 'n' representations, for this research work, we have defined four representations: C, SystemVerilog, Timed Automata (UPPAAL), and Meta-Model DSML. Each transformation is designed to preserve the semantics and structure of the source model while ensuring compatibility with the target language's syntax and constraints. Bidirectional transformations (BX) enable seamless conversion between different modeling languages, ensuring consistency and synchronization across representations. In the context of **Blended Modeling** and **Higher-Order Transformations (HOTs)**, transformation rules facilitate the interoperability between the four notations. These rules provide a structured approach to maintaining equivalence between abstract models and their concrete representations. However, defining such rules is highly challenging due to syntactic and semantic differences among these languages. This section formalizes the **bidirectional transformation rules**, highlighting key challenges and the innovative strategies used to address them.

A **bidirectional transformation rule** (BTR) ensures that a construct in source language *L1* can be mapped to a semantically equivalent construct in target language *L2* and vice versa. However, in practice, not all constructs have a one-to-one correspondence. To handle such cases, transformations involve:

- **Preservation:** Retaining core semantic meaning across transformations.
- **Workarounds:** Creating alternative representations where direct mappings do not exist.
- **Consistency Management:** Ensuring bidirectionality without information loss.
- **Automation Strategies:** Efficient parsing and transformation mechanisms.

This study systematically defines transformation rules across multiple modeling paradigms. These rules account for syntactic, structural, and semantic variations among these languages, aiming to preserve key functional properties during transformation.

**3.4.1 Transformation complexity and scope.** While this research involves four core notations, the true transformation space extends well beyond four simple conversions. Due to the bidirectional nature of the framework, each notation is capable of being transformed to and from every other notation. Consequently, the total number of directed transformations is significantly higher.

Let N be the number of distinct notations considered in the framework. The total number of directed transformations is given by:

$$T = N * (N - 1)$$

For : N = 4

$$T = 4 * (4 - 1) = 12$$

Thus, the proposed transformation framework realizes **12** unique bidirectional transformation paths (Fig 4), forming a complete directed graph among the four notations.

To maintain clarity and avoid redundancy, this transformation section focuses solely on the transformations originating from the C language to the other three notations. These include: SystemVerilog, UPPAAL (Timed Automata), and DSML. The remaining transformation paths (SystemVerilog Others, UPPAAL Others, DSML Others) follow structurally similar principles and are provided in the extended documentation..

The decision to focus on C-to-X transformations in this section stems from multiple factors. Firstly, C serves as a fundamental language for system modeling and embedded systems, making it a logical starting point for showcasing transformation methodologies. Secondly, including all bidirectional transformations within a single paper would introduce excessive complexity, making it difficult to maintain clarity and readability. By focusing on C-to-X transformations, this study provides a structured and comprehensible foundation while acknowledging the need for future extensions to fully formalize bidirectional mappings across all languages.

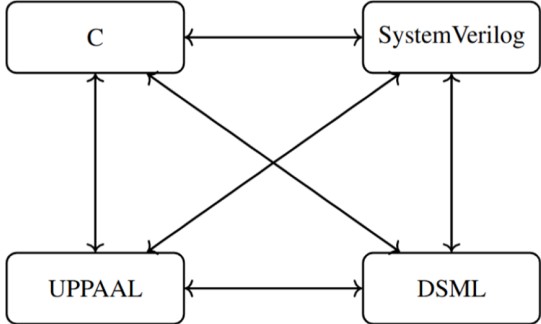

**Fig 4**. Directed transformation graph across multiple notations in embedded systems.

**3.4.2 C to timed automata (UPPAAL) transformation rules.** The transformation of **imperative C code** into **declarative Timed Automata (UPPAAL)** is a cornerstone of the blended modeling framework, enabling formal verification of real-time embedded systems. There were a lot of challenges while transforming C code to Timed Automata code. Timed Automata require all locations to be pre-defined before being used in transitions, which is not a requirement in C. The biggest challenge was to define all the locations before they are used between transitions. To resolve this, the entire C file is first parsed to count and store locations. Therefore, all locations are pre-created based on this count before transformation begins. Moreover, Uppaal lacks equivalent constructs for C Preprocessor directives. Thus, these directives are transformed into comments in the UPPAAL model. When transforming back, comments are reinterpreted as preprocessor directives. Another challenge was to handle *Enum*. UPPAAL does not support enum. Each typedef enum is converted into a set of constant integer variables:

```
typedef enum {light_on, light_off};
```

**Transforms into:**

```
const int light_on = 0;
```

```
const int light_off = 1;
```

Table 5 presents a structured transformation approach from C language constructs to their equivalent representations in UPPAAL. Each C construct, such as functions, control structures, loops, and data types, is systematically mapped to a corresponding UPPAAL representation. The transformation process addresses essential features like preprocessor directives, global variables, and function calls, adapting them within the constraints of UPPAAL's modeling semantics. Constructs that are incompatible or semantically complex, including pointers and extern variables, are either omitted or require restructuring. This mapping strategy establishes a structured approach for translating C programs into formal models compatible with UPPAAL verification.

**3.4.3 C to SystemVerilog transformation rules.** In our transformation framework, converting C code to SystemVerilog is achieved through a series of carefully defined rules that ensure semantic equivalence while accommodating inherent syntactic differences. Each C function is systematically transformed into a SystemVerilog module, encapsulated within a "`module VerilogSimulation; ... endmodule`" structure, so that every function corresponds to a specific hardware block. Variables in C are mapped to system variable data types based on their usage, ensuring that data storage and signal integrity are maintained in the hardware representation. Preprocessor directives such as `#include` and `#define` are transformed into constant declarations or preserved as comments, thereby retaining essential macro definitions and inclusion information. Additionally, `typedef enum` constructs in C are converted into SystemVerilog enumerations or constant definitions with distinct values to represent various states. Control flow constructs, including if-statements, switch-case structures, and loops, are encapsulated within "`begin ... end`" blocks, effectively modeling them as state machines to reflect the sequential and conditional behavior inherent in the source code. Function prototypes and definitions are similarly transformed by enclosing them within "`function ... endfunction`" constructs, ensuring that both the interface and the implementation details are preserved. These rules collectively address the challenges of mapping an imperative software language to a hardware description language, thereby enabling robust and semantically sound bidirectional transformations.

Table 6 provides a structured comparison between C and SystemVerilog constructs, detailing how various elements of C code are transformed into SystemVerilog. The first column lists common C constructs along with example code snippets, while the second column presents their corresponding representations in SystemVerilog. The third column outlines the transformation rules, ensuring a clear mapping between the two languages.

**Table 5**. C to uppaal transformation rules.

| C Representation | UPPAAL Representation | Transformation Rule |
|---|---|---|
| **Entire C Code Structure** Example: int main() { ... } | **Mapped to a UPPAAL Template** Example: <template>...</template> | The entire C code structure is transformed into a UPPAAL template, representing the main process of execution. |
| **Preprocessor Directives (#define, #include)** Example: #define MAX 10 #include <stdio.h> | **Defines mapped to const; includes omitted** Example: const int MAX = 10; | #define macros are converted into UPPAAL const declarations. #include directives are omitted since UPPAAL does not require them. |
| typedef enum Example: typedef enum {IDLE, RUNNING} State; | **Mapped to an Integer or UPPAAL typedef** Example: typedef int[0,1] State; | C enum is transformed into an integer range or a UPPAAL typedef, depending on its usage. |
| **Functions (definition & prototype)** Example: void foo() { x = 5; } | **Mapped to UPPAAL process or functions** Example: void foo() { x = 5; } | Functions in C are either represented as UPPAAL processes (if asynchronous) or as functions (if they return values). |
| **Control Statements (if, switch)** Example: if (x > 5) { y = 10; } | **Mapped to Guards in Transitions** Example: x > 5 -> y = 10; | Conditional statements are expressed as transition guards in UPPAAL. switch cases are transformed into multiple guarded transitions. |
| **Loops (for, while, do-while)** Example: for (int i = 0; i < 10; i++) { sum += i; } | **Mapped to UPPAAL Iterations using while** Example: while (i < 10) { sum += i; i++; } | Loops are represented as while constructs in UPPAAL, ensuring bounded execution. |
| **Variable Declarations (int, float, char)** Example: int count = 0; | **Mapped to UPPAAL int or clock** Example: int count = 0; | Integer and floating-point types are directly mapped, while UPPAAL clocks are used for time-dependent variables. |
| **Global Variables (extern)** Example: extern uint32_t count; | **Excluded in UPPAAL** | extern variables are not directly supported in UPPAAL and must be refactored into process-local or global declarations. |
| **Pointer Variables (*ptr)** Example: int *ptr; ptr = &var; | **Not Supported in UPPAAL** | UPPAAL does not support pointers; references must be handled via direct variable assignments. |
| **Structs (struct )** Example: struct {int a; float b;} data; | **Mapped to UPPAAL typedef struct** Example: typedef struct { int a; int b; } data_t; | Structs are converted into UPPAAL typedef struct, but floating-point members are transformed into integer equivalents. |
| **Function Calls** Example: foo(); | **Mapped to Function Calls or Synchronization** Example: foo(); | Function calls are retained if supported, otherwise modeled using templates |
| **Comments (//, /* */)** Example: // This is a comment | **Preserved as UPPAAL Comments** Example: // This is a comment | Comments are retained exactly as in C to ensure readability. |

Key transformations include encapsulating the entire C code structure within a SystemVerilog module, converting preprocessor directives into `localparam` or comments, mapping C functions to SystemVerilog `function` constructs, and transforming control statements and loops while maintaining block integrity. Additionally, data types such as `int`, `float`, and `struct` are carefully mapped to SystemVerilog equivalents. At the same time, certain constructs like `extern` variables and pointers require special handling or exclusion due to SystemVerilog's limitations. Table 6 serves as a reference guide for systematically converting C code into SystemVerilog, ensuring compatibility and functional equivalence.

**3.4.4 C to meta-model DSML transformation rules.** In the transformation from C to the DSML meta-model, various programming constructs are systematically mapped to corresponding DSML elements to ensure structural and semantic preservation. This transformation enables model-driven analysis and verification while maintaining the integrity of the original C code. The Table 7 outlines the transformation rules, illustrating how key C constructs such as functions, variables, control structures, and assignments are represented within the DSML framework. By encapsulating C elements into structured DSML components, the transformation facilitates interoperability with model-based tools, enabling further analysis, verification, and code generation.

Below is a concise yet comprehensive Table 7 summarizing the transformation rules from C to the Meta-Model DSML. This outlines how various C language constructs are mapped into DSML elements in our blended modeling framework:

Table 7 reflects our rigorous approach to mapping C constructs into DSML elements, ensuring that the semantic essence and structural integrity of the original C code are maintained within the DSML meta-model. The meta-model transformations serve as an intermediate representation between C, SystemVerilog, and Timed Automata. Constructs are

**Table 6**. C to SystemVerilog transformation rules.

| C Representation | SystemVerilog Representation | Transformation Rule |
|---|---|---|
| **Entire C Code Structure** Example: int main() { ... } | **Enclosed within module VerilogSimulation** Example: module VerilogSimulation; ... endmodule | The entire C code structure is encapsulated within a SystemVerilog module, ensuring that execution remains within Verilog's simulation scope. |
| **Preprocessor Directives (#define, #include)** Example: #define MAX 10 #include <stdio.h> | **Defines mapped to localparam; includes commented out** Example: localparam int MAX = 10; // #include <stdio.h> | #define macros are converted into localparam or const int declarations. #include directives are commented out since SystemVerilog does not require them. |
| **typedef enum** Example: typedef enum {IDLE, RUNNING} State; | **Mapped to SystemVerilog enum** Example: typedef enum logic [1:0] {IDLE, RUNNING} State; | C enum is transformed into a SystemVerilog enum with an explicit bit-width specification. |
| **Functions (definition & prototype)** Example: void foo() { x = 5; } | **Mapped to function ... endfunction** Example: function void foo(); x = 5; endfunction | Functions in C are directly mapped to SystemVerilog functions, preserving their parameter and return types. |
| **Control Statements (if, switch)** Example: if (x > 5) { y = 10; } | **Mapped to if ... else or case with begin ... end** Example: if (x > 5) begin y = 10; end | Conditional statements are enclosed within begin ... end to maintain block structure. switch statements are converted into case constructs. |
| **Loops (for, while, do-while)** Example: for (int i = 0; i < 10; i++) { sum += i; } | **Mapped to for ... end, while ... end constructs** Example: for (int i = 0; i < 10; i++) begin sum += i; end | Loops are directly mapped while ensuring begin ... end encapsulation. SystemVerilog does not support do-while, so it is rewritten as a while loop. |
| **Variable Declarations (int, float, char)** Example: int count = 0; | **Mapped to SystemVerilog data types (logic, int, real)** Example: int count = 0; | Integer and floating-point types are directly mapped, while pointers and arrays require transformation. |
| **Global Variables (extern)** Example: extern uint32_t count; | **Mapped to SystemVerilog import or extern** Example: extern int count; | Global extern variables are defined in one module and accessed using import or extern in SystemVerilog. |
| **Pointer Variables (*ptr)** Example: int *ptr; ptr = &var; | **Mapped to indirect memory references** Example: int ptr; ptr = var; | SystemVerilog lacks pointers; references are transformed into direct assignments where applicable. |
| Structs (struct) Example: struct {int a; float b;} data; | **Mapped to struct in SystemVerilog** Example: typedef struct {int a; real b;} data_t; | Structs in C are converted into SystemVerilog typedef struct. |
| **Function Calls** Example: foo(); | **Mapped to direct function invocations** Example: foo(); | Function calls are directly translated if arguments match SystemVerilog types. |
| **Comments (//, /* */)** Example: // This is a comment | **Preserved as-is in SystemVerilog** Example: // This is a comment | Comments are retained exactly as in C to ensure readability. |

mapped into XML-based representations that facilitate model-to-model transformations. The meta-model ensures that semantic correctness is maintained across conversions.

This shows how various C elements, such as preprocessor directives, functions, control statements, loops, and variable declarations, are transformed into equivalent DSML components. Each transformation rule ensures that the semantics of C code are preserved while adapting to the hierarchical and model-driven structure of DSML. The Table 7 highlights direct mappings, like functions becoming DSML operations and loops transforming into iteration nodes, as well as cases where C features, such as pointers, require alternative modeling approaches. This structured transformation is crucial for enabling systematic code translation and facilitating model-based analysis and verification.

In conclusion, this study establishes a structured framework for language transformation across C, SystemVerilog, UPPAAL, and a meta-model-based DSML. While the complete bidirectional transformation rules are extensive, this section presents only C-to-X transformations to provide a clear and foundational understanding of the methodology. The challenges and strategies outlined here highlight key aspects of preserving syntactic and semantic integrity during transformation.

**Table 7**. C to meta-model DSML transformation rules.

| C Representation | Meta-Model DSML Representation | Transformation Rule |
|---|---|---|
| **Entire C Code Structure** Example: int main() { ... } | **Mapped to a DSML Model Element** Example: <System name="MainSystem"> <Component name="MainComponent" /></System> | The entire C code structure is transformed into a root system element in DSML, containing components representing functions and execution units. |
| **Preprocessor Directives** (#define, #include) Example: #define MAX 10 #include <stdio.h> | **Mapped to Constants and Library References** Example: <Constant name="MAX" value="10" /> | #define macros are converted into DSML constants, while #include directives are transformed into library references where applicable. |
| typedef enum Example: typedef enum {IDLE, RUNNING} State; | **Mapped to an Enumeration in DSML** Example: <Enumeration name="State"><Literal name="IDLE" /><Literal name="RUNNING" /></Enumeration> | Enumerations in C are directly mapped to DSML enumeration types with corresponding literals. |
| **Functions (definition & prototype)** Example: void foo() { x = 5; } | **Mapped to DSML Operations within Components** Example: <Component name="Computation"><Operation name="foo"><Statement> x = 5; </Statement></Operation></Component> | Functions are represented as DSML operations contained within components. |
| **Control Statements** (**if, switch**) Example: if (x > 5) { y = 10; } | **Mapped to Conditional Nodes** Example: <Conditional condition="x > 5"><Assignment target="y" value="10" /></Conditional> | Conditional statements are modeled as conditional nodes in DSML with corresponding conditions and assignments. |
| **Loops** (**for, while, do-while**) Example: for (int i = 0; i < 10; i++) { sum += i; } | **Mapped to Iteration Nodes** Example: <Loop type="for" condition="i < 10"><Assignment target="sum" value="sum + i" /></Loop> | Loops are converted into iteration nodes, ensuring that loop behavior is captured in DSML. |
| **Variable Declarations** (**int, float, char**) Example: int count = 0; | **Mapped to DSML Attributes** Example: <Component name="DataStorage"><Attribute name="count" type="int" initialValue="0" /></Component> | Variables are transformed into DSML attributes within their respective components. |
| **Global Variables** (**extern**) Example: extern uint32_t count; | **Mapped to Shared Attributes** Example: <GlobalAttribute name="count" type="uint32" /> | Global variables are represented as shared attributes accessible across model elements. |
| **Pointer Variables** (*****ptr**) Example: int *ptr; ptr = &var; | **Not Supported Directly in DSML** | Pointers are not directly supported; instead, references are modeled using object relationships or ID-based mappings. |
| **Structs** (**struct**) Example: struct {int a; float b;} data; | **Mapped to DSML Composite Data Types** Example: <CompositeType name="data"><Attribute name="a" type="int" /><Attribute name="b" type="float" /></CompositeType> | Structs are transformed into composite types in DSML, maintaining their hierarchical structure. |
| **Function Calls** Example: foo(); | **Mapped to DSML Call Expressions** Example: <CallExpression operation="foo" /> | Function calls are retained as call expressions within DSML models. |
| **Comments** (**//, /* */**) Example: // This is a comment | **Preserved as DSML Annotations** Example: <Annotation text="This is a comment" /> | Comments are converted into DSML annotations to retain documentation within the model. |

## 3.5 Critical analysis of round-trip transformations in blended modeling

Blended modeling integrates multiple modeling paradigms by enabling transformations between 'n' different notations, such as C, SystemVerilog, Timed Automata (UPPAAL), and DSML meta-models. While these transformations facilitate interoperability, a critical challenge is ensuring bi-directionality, i.e., that transformations can be reversed without loss of information. An ideal transformation framework should allow for **round-trip consistency**, meaning that translating a model from one language (**L1**) to another (**L2**) and then back (**L1 → L2 → L1**) should **reproduce the exact source representation**. However, due to **semantic mismatches** between languages, this is not always achievable. Each of the target languages, including C, SystemVerilog, Timed Automata, and DSML, has different abstractions and expressiveness, making certain constructs non-reversible. The key challenge is designing transformation rules that minimize information

loss, ensuring that the original structure can be reconstructed as accurately as possible when transforming back. Achieving full bi-directional fidelity across these languages remains an open challenge requiring traceability mechanisms, hybrid modeling approaches, and improved transformation heuristics.

In model-driven transformations, ensuring a lossless round-trip conversion between different languages is a crucial challenge. Fig 5 illustrates round-trip transformations across multiple notations (1 to n) in a blended modeling framework. Each forward transformation introduces some **delta loss**, indicating potential information loss. The return transformation (n → 1) may not fully restore the original due to accumulated losses. It highlights the challenge of achieving accurate, lossless bidirectional transformations. The primary goal of round-trip transformations (L1 → L2 → L1) is to maintain semantic integrity, meaning that converting a source language (L1) to a target language (L2) and then back to the source should ideally yield the original code. However, due to differences in language constructs, information loss can occur. Table 8 highlights some key challenges encountered during bidirectional transformations between C, SystemVerilog, UPPAAL, and DSML, along with the strategies adopted to mitigate these losses.

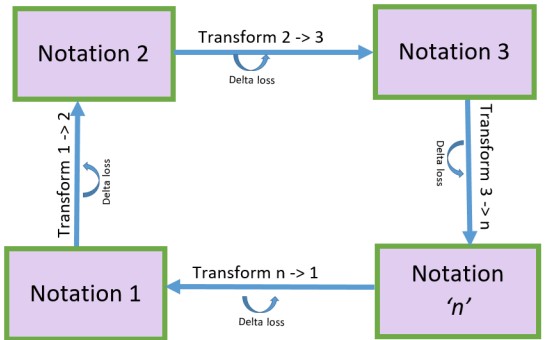

**Fig 5**. **Round-trip transformations across multiple notations.**

**Table 8**. **Round trip transformations analysis.**

| Round Trip Transformation | Challenge | Adopted Strategy | Information Loss |
|---|---|---|---|
| C → SystemVerilog → C | *extern* variable declaration in C has no equivalent in SystemVerilog. | extern is lost in transformation and cannot be restored. | Yes |
| C → SystemVerilog → C | *uint8_t* and *uint32_t* has no equivalent in SystemVerilog. | During round trip, all int types map back to int in C, losing original type precision. | Yes |
| C → SystemVerilog → C | In C, array is defined as *txx[13]* and in SystemVerilog, it is defined as *txx[12:0]*. | Handled programmatically to ensure correct transformation back to C. | No |
| C → SystemVerilog → C | *#include* directives have no direct mapping in SystemVerilog. | Preserved as comments in SystemVerilog, parsed, and restored in C. | No |
| C → UPPAAL → C | *#define* preprocessor directives have no equivalent in UPPAAL. | Transformed into const int in UPPAAL and remains const int on round trip back to C. | Yes |
| C → SystemVerilog → C | *#define* preprocessor directives have no equivalent in SystemVerilog. | Transformed into const int in SystemVerilog and remains const int on round trip back to C. | Yes |
| C → UPPAAL → C | *#include* directives have no direct mapping in UPPAAL. | Preserved as comments in UPPAAL, parsed, and restored in C. | No |
| C → UPPAAL → C | *enum* types from C has no equivalent in UPPAAL. | On round trip back to C, enum types remain const int. | Yes |
| C → UPPAAL → C | *Print* statements from C have no direct equivalent in UPPAAL. | Preserved in UPPAAL comments, parsed, and restored in C. | No |
| C → UPPAAL → C | *Function calls* from C are transformed into transitions in UPPAAL, as no equivalent is available. | Handled programmatically to ensure function calls are correctly restored in C. | No |

While the Table 8 also outlines some of the most significant challenges and solutions in round-trip transformations, there are numerous other cases where discrepancies may arise due to language-specific constructs. Ensuring complete bidirectional transformation remains an ongoing research challenge in blended modeling, requiring further refinements and adaptive strategies to preserve as much information as possible.

### 3.6 Pseudocode (Algorithm)

This section provides the pseudocode for the bidirectional transformation engine, a system designed to enable the seamless translation of code between multiple programming languages. The engine ensures that updates made in one language are automatically reflected in the corresponding target languages, maintaining consistency across all representations. Through a series of structured steps, the engine saves updated code, executes transformations for other languages, detects errors, and refreshes the graphical user interface (GUI) to display the most current code. The pseudocode detailed in this section illustrates the core workflow of the engine, emphasizing its ability to manage language synchronization and ensure the integrity of transformations across diverse codebases. The pseudocode is as follows:

**Algorithm 1.   Pseudocode for blended modeling framework.**

```
Initialization Phase: Initialize C, SystemVerilog, Timed Automata, and Meta-Model DSML tree.
On User Update:
  1. User modifies code and triggers save.
  2. Retrieve the source file and save the updated code:[1]
        source_file ← GetSourceFile(updated_language)
        SaveToFile(source_file, updated_code)

Transformation Phase: for each target_language in {C, SystemVerilog, UPPAAL, DSML} do
    if target_language ≠ updated_language then
        Lexical Analysis: [1] lexer ← InitializeLexer(source_code)
        token_stream ← GenerateTokens(lexer)

        Parsing: [1] parser ← InitializeParser(token_stream)
        parse_tree ← parser.StartRule()

        Error Handling: [1] if ParseErrorsExist(parser) then
          └ THROW "Syntax Error in Source Code"
        AST Construction: [1] ast ← ConvertParseTreeToAST(parse_tree)

        Code Transformation: [1]
        visitor ← InitializeVisitor(source_language, target_language)
        transformed_code ← "" for each node in ast do

        transformed_code ←
        transformed_code + visitor.ApplyTransformation(node)
        Save Transformed Code: [1]
        └ SaveToFile(GetSourceFile(target_language), transformed_code)
Update Phase: for each language in {C, SystemVerilog, UPPAAL, DSML} do
    [1] source_file ← GetSourceFile(language)
    latest_code ← ReadFromFile(source_file)
    UpdateGUI(language, latest_code)
repeat
until all GUI updates complete;
```

The **Bidirectional Transformation Engine** is designed to facilitate seamless code transformation between **C, SystemVerilog, UPPAAL (Timed Automata), and Meta-Model DSML**. One can modify any of these representations via a **GUI**, which displays all four languages in separate tabs, each linked to an underlying **source file**. Upon pressing the **Save**

button, the system determines which language was last updated and commits the changes to its respective source file. The engine then executes transformations for the other three languages by leveraging **ANTLR's parsing and transformation process**.

The transformation process begins with **lexical analysis**, where the **lexer** tokenizes the source code into meaningful components. The **parser** then applies grammar rules to construct a **parse tree**, ensuring the correctness of the input. If any **syntax errors** are detected, the process is halted, and an error is thrown. Upon successful parsing, an **Abstract Syntax Tree (AST)** is generated, which serves as the structured representation of the code. The **visitor pattern** is used to traverse this AST, applying **language-specific transformation rules**. Each node in the AST is processed, and the transformed code is iteratively constructed and saved to the corresponding **source file** for the target language.

This approach ensures **consistency** among the representations, enabling bidirectional transformations while preserving **structural integrity**. Although some **information loss** is inevitable due to semantic differences among languages, careful preservation strategies (such as maintaining preprocessor directives in comments) mitigate these challenges. This transformation engine allows for an efficient and **automated round-trip engineering process**, which is crucial for model-driven development, embedded system design, and formal verification workflows.

This pseudocode effectively captures the bidirectional transformation process, ensuring consistency and accurate synchronization of code across multiple languages. The integration of parsing, structured error handling, and the visitor design pattern establishes a reliable framework that facilitates language transformations while maintaining both syntactic accuracy and semantic consistency.

## 4 Implementation architecture

The framework is equipped with an interactive graphical user interface developed using Java Swing, integrated within the Eclipse IDE to benefit from its robust development environment. This interface facilitates real-time transformations among multiple language representations. While the current implementation supports C, SystemVerilog, Timed Automata (UPPAAL), and DSML, the underlying architecture is designed to be extensible, allowing integration of additional notations as needed for broader applicability.

Eclipse was chosen as the primary development environment for its robust support for modular development, debugging, and integration of various frameworks. The Eclipse Modeling Framework (EMF) was utilized to define and manage the Meta-Model for the DSML representation. Additionally, the Eclipse environment facilitated the organization of transformation rules, grammar files, and GUI (Graphical User Interface) components into distinct packages, ensuring a clean and scalable architecture.

Fig 6 presents the implementation architecture of the proposed blended modeling framework for the design and verification of safety-critical embedded systems. The architecture begins with the grammars of supported languages, C, SystemVerilog (SV), Timed Automata (TA), and DSML, notated on the left. Each grammar is processed through a lexer and parser, generating an Abstract Syntax Tree (AST) for each respective notation. These ASTs are maintained in a centralized repository and serve as the foundation for transformation. The visitor pattern is employed to traverse the AST tokens, applying notation-specific transformation rules. These transformations are governed by a command design pattern that decouples the transformation logic and allows modular execution. The output is then propagated to update the other modeling notations consistently, preserving synchronization across all views. This layered and modular structure ensures scalability, enabling the seamless integration of additional notations or transformation rules in the future, while maintaining the system's responsiveness and maintainability.

### 4.1 Modular development approach (BackEnd)

To maintain clarity and scalability, the GUI and transformation logic were implemented using a modular approach. The system adopts a **layered architecture** to decouple the GUI, transformations, and utilities:

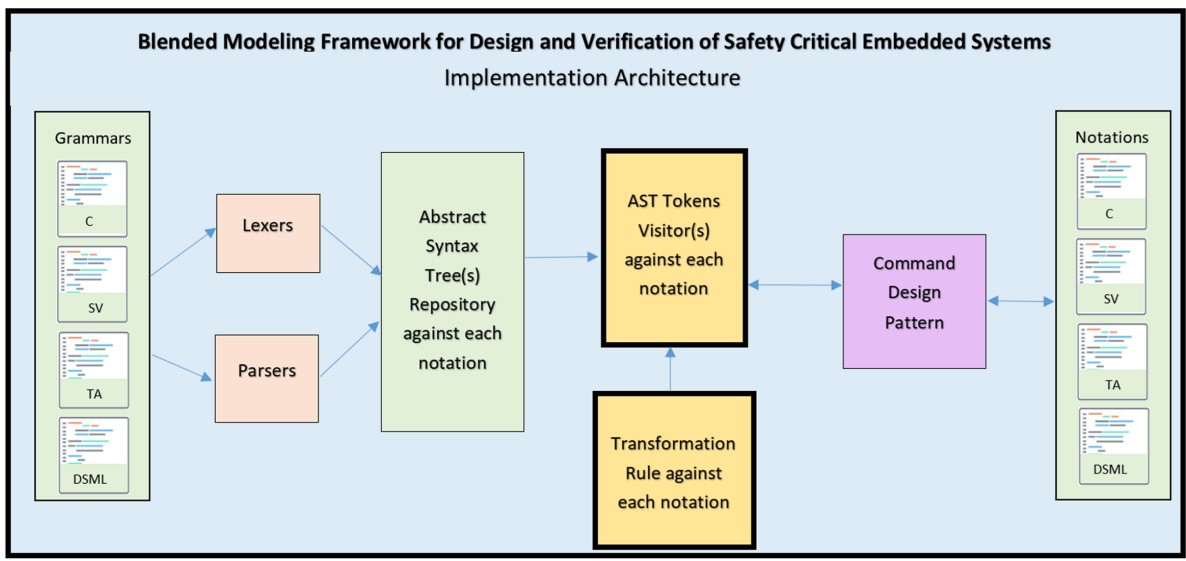

**Fig 6**. Implementation architecture of the proposed blended modeling framework.

**Separate Packages for Each Transformation**:

- **CTransformations**: Contains ANTLR grammar files, visitor classes, and utility methods specific to transforming C representations to other formats.
- **SVTransformations**: Handles transformations from SystemVerilog to other representations.
- **TATransformations**: Encapsulates rules and logic for converting Timed Automata to other formats.
- **DSMLTransformations**: Includes ANTLR grammar files, visitor classes, and utility methods for DSML-to-other transformations.
- **GUIComponents**: Contains all GUI-related components, including tab management, event handling, and UI rendering.
- **Utilities**: Includes shared utilities for file handling, logging, and notifications.

This modular approach ensures that any updates or enhancements to a specific transformation do not impact the rest of the application, promoting maintainability and extensibility. The abstract syntax meta-model for the DSML representation was developed using the EMF DSML framework, with dsml.ecore defines the meta-model in Ecore. This meta-model serves as the foundation for creating instances that represent various DSML elements.

## 4.2 GUI design and functionality (FrontEnd)

GUI features a user-friendly interface with four tabs arranged side by side, corresponding to the four representations (Fig 7). Each tab allows one to view, edit, and manage the respective representation. Changes made in one tab can be propagated to other representations through real-time transformations.

The GUI implementation is structured around a multi-tab layout that displays all four modeling representations simultaneously, facilitating effortless navigation between C, SystemVerilog, Timed Automata, and DSML views. Transformations are seamlessly handled through a central transformation engine, triggered explicitly upon detecting code changes. The system leverages ANTLR grammars and tailored visitor implementations to parse and convert code efficiently, ensuring

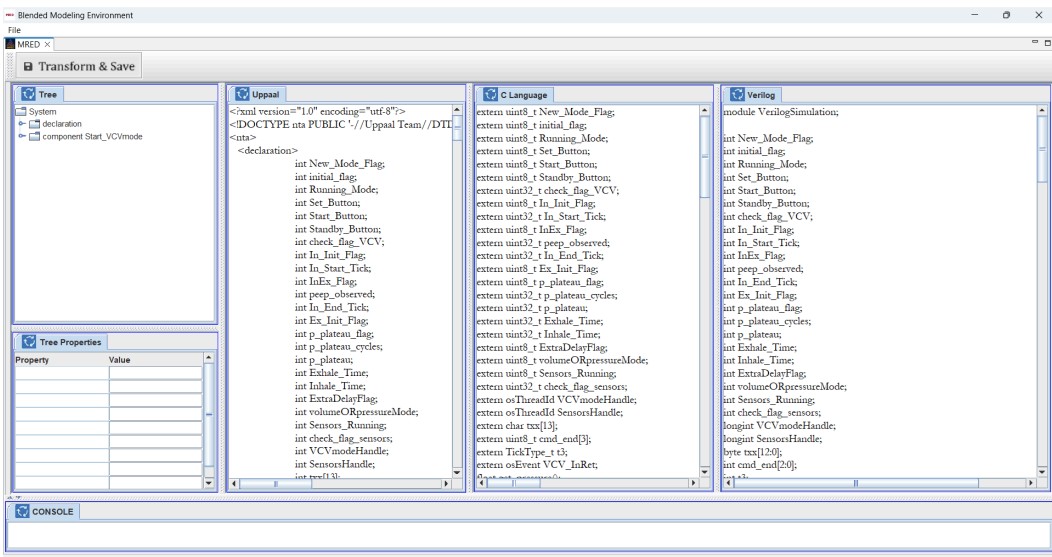

**Fig 7**. Graphical User Interface (GUI) of the proposed blended modeling framework.

real-time synchronization across all views. Background transformations enable to continue working in one tab while transformations are applied to others, without experiencing any performance lag. Furthermore, the application supports comprehensive file management features, allowing import and export of models in various formats. Changes are persistently stored within the workspace, maintaining data consistency and ensuring smooth integration into the modeling workflow.

The blended modeling framework features an interactive GUI that enables seamless real-time transformation across multiple modeling notations. As code is modified within the active tab, the system detects changes upon request and triggers the transformation process. A live parsing engine processes the updated code to generate the abstract syntax tree (AST), and transformation rules, implemented using the visitor pattern, are applied to produce corresponding representations across the remaining views (C, SystemVerilog, Timed Automata, or DSML). These representations are updated instantly and synchronized across tabs without requiring manual intervention. The GUI, developed using Java Swing integrated with Eclipse RCP, provides an intuitive and responsive environment for model management. Its modular and extensible architecture allows easy incorporation of additional languages and representations, supporting future scalability. This design simplifies complex model handling and makes the framework accessible with limited expertise in formal methods.

To promote reproducibility and facilitate future research, the complete implementation of the proposed blended modeling framework has been made publicly available. The source code, including grammars, transformation rules, visitor classes, and the Eclipse RCP-based GUI, is hosted on GitHub [35]. Additionally, a pre-built Eclipse RCP product is also provided [35] for direct execution without requiring compilation. The repository contains detailed instructions for setup and usage. Researchers and practitioners are encouraged to explore, reuse, or extend the tool to support additional languages or verification workflows in the embedded systems domain.

## 5 Proof of concept - validation and evaluation

Two case studies have been conducted to validate the proposed bi-directional transformation framework: (1) a ventilator system and (2) a cruise control system. These case studies demonstrate how transformations are applied in real-world embedded systems, where different aspects of the system are modeled using multiple representations. The ventilator

system case study consists of two use cases, each highlighting different subsets of the specified languages, ensuring a thorough evaluation of the transformation process.

The transformation engine operates in a dynamic manner where the last modified representation is identified as the source language, while the other three representations are transformed accordingly. The transformed representations are then seamlessly reflected in the GUI, ensuring real-time synchronization between all four representations.

## 5.1 Case study 1: Ventilator system

**5.1.1 Overview of ventilator systems in embedded design.** A ventilator is a **real-time embedded system** used in medical applications to regulate a patient's breathing cycle. The system automates the inhalation and exhalation process by controlling airflow based on parameters such as pressure, volume, and timing, as presented in Fig 8. The software governing the ventilator operates under strict **real-time constraints**, ensuring precise breath delivery while continuously monitoring **sensor data** for feedback. The embedded system comprises **actuators to control valves, RTOS-based task scheduling for managing multiple breathing modes**, and **safety mechanisms to handle system failures**. Given its complexity, a ventilator system is an ideal case study to validate the **multi-representational transformation framework**, as it encompasses both **software and hardware-level descriptions.**

**5.1.2 Use Case 1: Start ventilator mode.** This use case illustrates the ventilator's transition into its active mode of operation, where all essential control parameters are verified and configured before ventilation begins. The transformation framework ensures that changes made at this stage are consistently reflected across all four representations, C, SystemVerilog, Timed Automata (UPPAAL), and the DSML model, maintaining semantic alignment.

At the core of this scenario is the initialization phase, which is triggered either when the ventilator is powered on or when switching between operating modes. The focus here is to confirm that the system is prepared for safe operation.

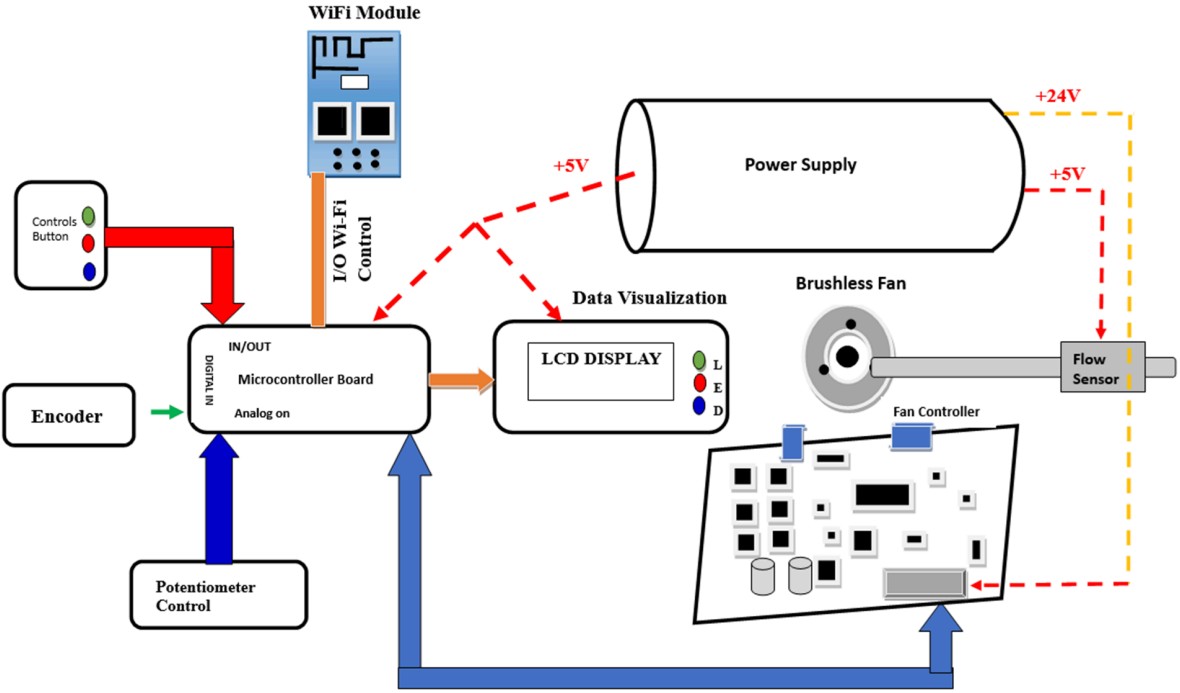

**Fig 8**. **Abstract model of the ventilator system.**

Key tasks include initializing control variables, deactivating non-critical threads to reduce startup overhead, using system timers for task scheduling, and handling exceptional cases such as sensor readiness or calibration checks.

This scenario also serves as a functional test case for the transformation process, given its variety of programming features commonly used in embedded systems. It integrates basic and advanced elements such as fixed-width data types (`uint8_t`, `uint32_t`), function declarations and invocations, conditional and loop structures, and various assignment operations. These constructs are transformed across the target languages while preserving structural and semantic integrity. For example, fixed-width integers in C are mapped to type-compatible constructs in SystemVerilog, abstract clocks or variables in UPPAAL, and corresponding elements in the DSML meta-model, supporting a reliable and synchronized multi-notation representation. The Transformations work as follows:

**Transformation Rule: TR-1.1**

```
C: if(New_Mode_Flag)
    {
    New_Mode_Flag = 0;
    }

SystemVerilog: if (New_Mode_Flag) begin
            New_Mode_Flag = 0;
                end

Uppaal:<transition>
        <source ref="id0"/>
        <target ref="id1"/>
        <label kind="guard">
            New_Mode_Flag
        </label>
        <label kind="assignment">
                New_Mode_Flag = 0
        </label>
      </transition>

DSML: <statement>
        <if
        keyword="if"
        variable="New_Mode_Flag"
        operator=""
        value="">
        <assignment
        leftAttribute="New_Mode_Flag"
        operator="="
        AssignedSide="0"/>
        </if>
    </statement>
```

Function prototypes and function calls play a crucial role in this case, as they define modular operations within the ventilator system. The transformation framework ensures that function signatures remain intact across all representations, maintaining logical consistency. Control flow statements, particularly if-else constructs, regulate ventilator operations based on sensor inputs and system states. These are transformed into guarded transitions in UPPAAL (TR-1.1), structured begin-end conditionals in SystemVerilog, and structured decision blocks in DSML. Additionally, the case includes various types of assignments such as constant assignments, variable-to-variable assignments, function call-based assignments, and structured array assignments. These assignments are systematically transformed, preserving semantic correctness across all representations.

When one modifies any of the four representations, an Abstract Syntax Tree (AST) is generated, which serves as an intermediate representation for transformation. The visitor-based transformation logic then traverses the AST, applying the pre-defined transformation rules. For instance, in SystemVerilog, extern uint8_t from C is transformed into an int mode_flag; whereas in UPPAAL, an if statement from C is translated into a state transition with an associated guard condition. The DSML representation, on the other hand, structures control flow statements as hierarchical XML-based decision blocks.

When transforming from **C to SystemVerilog**, the `extern` keyword is omitted, and `uint8_t` is converted into `int`, ensuring compatibility with SystemVerilog's data type system. Similarly, for **C to Timed Automata (UPPAAL) transformations**, `uint8_t` and `uint32_t` are mapped to `int`, maintaining numerical consistency. However, in the case of **C to Meta-Model DSML**, the **data type syntax and semantics are preserved**, ensuring that the high-level abstraction retains the original structure of the embedded system representation.

The transformation of arrays across different representations is also managed. In SystemVerilog, arrays are transformed into the format `byte txx [12:0]`, while in C, they are represented as `extern char txx[13]`. During transformation from C to Timed Automata (UPPAAL), the `char` data type is mapped to `string`, whereas the array structure `txx[13]` remains unchanged. Finally, in the Meta-Model DSML representation, both the syntax and semantics of the array are preserved without modification, ensuring consistency in the structural representation across transformations.

Comments remain unchanged during transformations between C and SystemVerilog, preserving both single-line and multi-line formats. However, Timed Automata (UPPAAL) follows a different comment syntax, using `<!-- -->`, requiring conversion during transformation. Despite this syntactic adjustment, no information is lost. In contrast, the Meta-Model DSML representation maintains comments in their original source language syntax, ensuring that annotations remain intact across transformations.

This use case demonstrates the effectiveness of the proposed transformation framework in managing multi-paradigm embedded system representations while preserving logical and functional equivalence.

**5.1.3 Use Case 2: Mode shift control.** This use case addresses the controlled switching between various ventilation modes, such as Pressure-Controlled (PCV), Volume-Controlled (VCV), and Assisted-Controlled (ACV). The system handles these transitions by updating control parameters and suspending or resuming relevant threads, ensuring no disruption in ongoing operations.

The mode switching logic is implemented through a `switch` statement based on the `Running_Mode` variable, covering all defined modes with a `default` case for safety. The transformation framework accurately maps these constructs into C, SystemVerilog, Timed Automata, and DSML representations, preserving the structure and logic essential for system stability.

The implementation of this use case includes several key programming constructs that are transformed into equivalent representations across different modeling languages.

The *extern* variable declarations are essential for storing key system parameters. The ventilator system relies on multiple `extern` variables of type `uint8_t` and `osThreadId`. These include critical parameters such as `Running_Mode`, ventilation control flags, and thread handles for different ventilation modes. In SystemVerilog, the `extern` keyword is

removed as there is no equivalent construct, hence information loss on round-trip transformation, while `uint8_t` is transformed into `int`. For Timed Automata (UPPAAL), `osThreadId` is converted into an integer representation to align with the automaton-based execution model.

The *switch-case* structure plays a crucial role in handling mode transitions. The `switch` construct determines the current mode of operation by evaluating `Running_Mode`. Each case represents a specific ventilation mode, ensuring smooth transitions between different states. Within each case, multiple assignment statements update system parameters dynamically. These include updates to control variables such as `volumeORpressureMode`, which dictates whether the ventilator operates in volume or pressure control mode. Additionally, function calls are executed to apply necessary system adjustments, including `Update_Internal_Parameters()` and `osThreadSuspend()`, which dynamically suspend or resume ventilation threads. The inclusion of comment statements provides clarity regarding system behavior, helping in debugging and traceability. To maintain execution flow integrity, the `break` statement is used at the end of each case, ensuring that only one case executes at a time and preventing unintended fall-through behavior.

**Transformation Rule: TR-2.1**

```
C:  switch(Running_Mode)
 {
  case 1:
   {
//other code
break;
   }
   //rest of the cases and code
 }
```

---

```
SystemVerilog:
case (Running_Mode)
1: begin
  //other code
end
//rest of the cases and code
```

---

```
Uppaal:
<transition>
  <source ref="id0"/>
  <target ref="id1"/>
  <label kind="guard">
    switch== Running_Mode
  </label>
</transition>
<transition>
  <source ref="id1"/>
```

```
  <target ref="id2"/>
  <label kind="guard">
     Running_Mode==1
  </label>
   //other code
</transition>
//BREAK_STATEMENT
```

---

```
DSML:
<switch
keyword="switch"
value="Running_Mode">
<switchcase
keyword="case"
value="1">
//other code
</switchcase>
//rest of the cases and code
```

---

Regarding **transformations across representations**, the proposed framework ensures structural preservation and semantic correctness. In **SystemVerilog**, the `switch` construct is directly mapped, maintaining the same logical structure (TR-2.1). Assignments, function calls, and comments are transformed into their equivalent SystemVerilog constructs while preserving their execution order. For **Timed Automata** (**UPPAAL**), each `case` is mapped to a transition within the automaton model. Assignment statements are transformed into transition assignments (`kind=assignment`), while function calls are handled as either `kind=guard` or `kind=assignment`, depending on whether they act as conditions or actions in the state transitions. Finally, in **Meta-Model DSML**, the `switch` structure and its cases are represented in XML format, with `<switch>` and `<switchcase>` elements encapsulating assignments, function calls, and comments in a hierarchical manner.

The proposed framework ensures that complex mode-switching remains structurally intact across all representations while transforming each construct to the target language. The bidirectional transformation approach guarantees that modifications in one representation are transformed into all others, thereby enabling seamless verification of ventilator system behavior.

### 5.2 Case Study 2: Cruise control system

**5.2.1 Overview of cruise control systems in embedded design.**  The cruise control system operates through real-time interactions between software and hardware components (Fig 9). It maintains a steady vehicle speed by automatically adjusting the throttle based on feedback from speed sensors. The system allows the driver to set, increase, decrease, or cancel the desired speed through button interactions. The embedded control logic ensures that speed remains within predefined limits while adjusting for variations in terrain and resistance.

**5.2.2 Use Case 1: Speed monitoring.**  The execution of the cruise control use case involves continuous monitoring of vehicle speed and dynamic state transitions based on inputs. The system initializes in the `CRUISE_OFF` state, where manual speed control is required. Upon activation, the system transitions to `CRUISE_ON`, where it maintains a

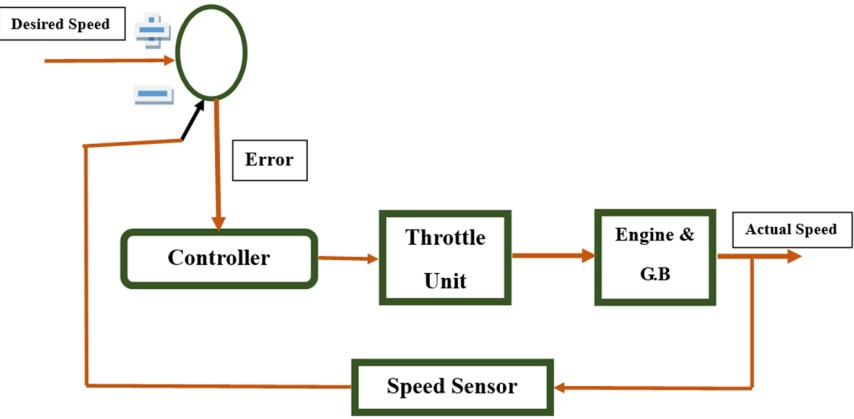

**Fig 9**. **Abstract model of the automotive cruise control system.**

steady speed using the `controlThrottle()` function. If the driver manually adjusts the speed, the system enters the `CRUISE_HOLD` state, allowing gradual acceleration or deceleration while preserving automated control. These transitions rely on a structured implementation using enumerations, conditional logic, function calls, and iterative loops, ensuring precise control over vehicle speed regulation. Applying transformation rules across different representations is crucial to maintaining the integrity and correctness of the cruise control system.

The cruise control system case study implementation incorporates a variety of fundamental programming constructs that are essential for embedded systems development. Preprocessor directives such as `#include` and `#define` are utilized to include standard input-output libraries for debugging and to define constants for speed limits, throttle control, and step increments. The system employs an enumeration (`typedef enum`) to define three operational states—`CRUISE_OFF`, `CRUISE_ON`, and `CRUISE_HOLD`—which govern the behavior of the cruise control logic. Global variables, including `currentSpeed`, `desiredSpeed`, `throttle`, and `state`, are used to maintain the system's state and dynamically update parameters based on real-time inputs.

Function implementations play a crucial role in executing the system's logic. The `readCurrentSpeed()` function simulates sensor data retrieval, while `controlThrottle()` adjusts the throttle based on speed differences to maintain the set speed. The `updateCruiseControl()` function incorporates a `switch` statement that processes inputs to set, change, or cancel cruise control, ensuring smooth state transitions. Conditional statements (`if-else`) are extensively used throughout the program to facilitate decision-making, such as adjusting speed limits, regulating the throttle, and handling system transitions. A `for` loop within the `main()` function simulates continuous monitoring of speed adjustments and system responses.

Additionally, `printf` statements provide real-time feedback, assisting in debugging and interaction by displaying speed, state transitions, and throttle adjustments. This combination of constructs ensures that the cruise control system functions effectively, simulating real-world embedded control mechanisms while enabling seamless transformations between different representations.

The transformation process follows a structured approach to ensure the correct mapping of constructs across different representations. Preprocessor directivessuch as `#include` and `#define` in C do not have direct equivalents in UPPAAL and SystemVerilog; therefore, they are transformed into comments and, on round-trip transformations, will be transformed back. The `typedef enum` construct, which represents cruise control states, is transformed (TR-3.1) into integer constants in both UPPAAL, as there is no equivalent available. However, in SystemVerilog, it is transformed into an SV syntax enum. Finally, the enum is transformed into a relevant tag in DSML.

**Transformation Rule: TR-3.1**

```
C:
typedef enum {
    CRUISE_OFF,
    CRUISE_ON,
    CRUISE_HOLD
} CruiseState;
CruiseState state = CRUISE_OFF;
```

---

```
SystemVerilog:
typedef enum logic[1:0] {
    CRUISE_OFF,
    CRUISE_ON,
    CRUISE_HOLD
 } CruiseState;
 CruiseState state=CRUISE_OFF;
```

---

```
Uppaal:
const int CRUISE_OFF = 0;
const int CRUISE_ON = 1;
const int CRUISE_HOLD = 2;
int state=CRUISE_OFF;
```

---

```
DSML:
<declaration>
<enum enumID="CruiseState">
<enumlist enumItem="CRUISE_OFF"/>
<enumlist enumItem="CRUISE_ON"/>
<enumlist enumItem="CRUISE_HOLD"/>
</enum>
<attribute
isGlobal=""
dataType="CruiseState"
name="state"
size=""
initialValue="CRUISE_OFF"/>
```

---

In C, a Function definition has a defined syntax with a return type, parameters, and code inside parentheses. In SystemVerilog, it is defined within `function...endfunction` keywords, while in UPPAAL, they are abstracted as templates or transitions depending on their role in the execution flow. Finally, DSML has relevant tags for each construct of concrete notations.

Control flow statements, including `if-else` and `switch-case`, are transformed into equivalent constructs across representations. In SystemVerilog, they are enclosed within `begin...end` blocks, while in UPPAAL, they are represented as guarded transitions between states. Loop constructs (TR-3.2), such as `for` loops, are converted into iterative transition structures in UPPAAL, preserving the logic while adapting to the target model's syntax. Loop start and end are preserved in Uppaal by introducing comment statements *//FOR-LOOP-OPEN* and *//FOR-LOOP-CLOSE* as it will preserve the opening and closing of the loop during the round-trip transformation cycle.

**Transformation Rule: TR-3.2**

```
C:
for (int i = 0; i < 20; i++) {
    // some code
}
```

---

```
SystemVerilog:
for (int i = 0;i < 20;i++) begin
      // some code
end
```

---

```
Uppaal:
//FOR_LOOP_OPEN
<transition>
   <source ref="id14"/>
   <target ref="id15"/>
   <label kind="assignment">j=0</label>
</transition>
<transition>
    <source ref="id14"/>
    <target ref="id16"/>
    <label kind="guard">
        j<1000000
    </label>
    <label kind="assignment">j++</label>
</transition>
<transition>
   <source ref="id16"/>
   <target ref="id15"/>
</transition>
```

```
//FOR_LOOP_CLOSE
// some code
```

---

```
DSML:
<statement>
<loop
type="for"
initial="int i=0"
condition="i < 20"
increment="i++">
//some code
</loop>
</statement>
```

---

Through this structured transformation approach, the cruise control system is effectively mapped across different representations, ensuring that its functional behavior remains intact while adapting to the syntactic and semantic constraints of each target language.

The ventilator system case study successfully validates the proposed transformation framework by demonstrating that a complex real-time embedded system can be effectively represented and synchronized across C, SystemVerilog, UPPAAL, and DSML. The study highlights how **different programming constructs**, including **conditional logic, function calls, and switch-case statements**, are accurately mapped to their corresponding representations while preserving semantic integrity.

## 6 Performance evaluation

To validate the efficacy of the proposed transformation framework, we conducted a comprehensive performance evaluation across all transformation pathways among the four modeling notations: C, SystemVerilog, UPPAAL, and DSML. The proposed transformation framework is evaluated based on multiple performance metrics, including transformation latency, round-trip accuracy, edge-case handling, memory usage, and scalability. These metrics were measured on a Windows 11 (64-bit) machine with an Intel Core i5-1135G7 processor (2.42 GHz) and 8GB RAM, ensuring realistic performance expectations.

### 6.1 Performance evaluation parameters

**1. Transformation Latency** Time required to convert source code to a target representation (measured for 100 lines of code or model elements). Results are summarized in Table 9. Transformation times vary due to differences in language complexity and the computational effort required to map constructs between source and target notations. For instance, transformations involving **UPPAAL** exhibit higher latency (110–130 ms for 100 LOC) because timed automata require flattening of state-space models and handling synchronization semantics. In contrast, **DSML-based transformations** are faster (60–70 ms) owing to high-level template-based abstraction and simpler structural mappings. Procedural languages like **C and SystemVerilog** show intermediate latency (80–100 ms), reflecting the moderate parsing and traversal costs associated with their control-flow constructs and data types.

**Table 9**. Transformation latency across different paths.

| Source → Target | Latency (ms) |
|---|---|
| C → UPPAAL | 120 |
| C → SystemVerilog | 80 |
| C → DSML | 100 |
| SystemVerilog → C | 90 |
| SystemVerilog → UPPAAL | 110 |
| SystemVerilog → DSML | 70 |
| UPPAAL → C | 130 |
| UPPAAL → SystemVerilog | 100 |
| UPPAAL → DSML | 85 |
| DSML → C | 95 |
| DSML → SystemVerilog | 60 |
| DSML → UPPAAL | 70 |

**2. Round-Trip Accuracy** Semantic fidelity is maintained during forward and reverse transformations and assessed through automated trace comparison and manual inspection. Results are shown in Table 10. Accuracy is largely determined by semantic alignment between source and target languages. Transformations between **C and SystemVerilog** achieve the highest fidelity (∼98%) due to structural similarity and well-defined control constructs. Lower accuracy (∼90%) is observed for paths involving **UPPAAL**, mainly because timed behaviors and preemption semantics are approximated during bidirectional mapping. Minor deviations also arise from **language-specific features**, such as preprocessor directives in C, concurrency constructs in SystemVerilog, or complex state hierarchies in UPPAAL.

**3. Edge-Case Handling** Success rate in managing syntactic and semantic complexities like recursion, nested constructs, and mixed data types. Results are presented in Table 11. The framework demonstrates robust handling of standard control constructs (100% success for switch statements across all languages). Variability in handling loops and mixed data types (80–95%) reflects challenges in mapping nested or heterogeneous structures while maintaining semantic fidelity. Higher success rates in DSML (90–95%) are due to its abstract, template-driven design, which accommodates diverse patterns more flexibly. Procedural languages, such as UPPAAL and C, require explicit handling of recursion, loops, and type heterogeneity, accounting for slightly lower success percentages.

**4. Memory Usage** *Average memory consumption during transformation.* Results are summarized in Table 12. Memory requirements are determined by AST sizes, internal DSML meta-model representations, and intermediate transformation

**Table 10**. Round-trip transformation accuracy and observations.

| Transformation Path | Accuracy (%) | Notable Causes of Deviation |
|---|---|---|
| C ↔ UPPAAL | 92 | Limited support for preprocessor directives |
| C ↔ SystemVerilog | 98 | Concurrency semantic misalignments |
| C ↔ DSML | 93 | Inconsistencies in variable naming and hierarchy |
| SystemVerilog ↔ C | 95 | Manual annotations needed for process constructs |
| SystemVerilog ↔ UPPAAL | 90 | Timing abstraction inaccuracies |
| SystemVerilog ↔ DSML | 95 | Structural match well maintained |
| UPPAAL ↔ C | 90 | Flattening of state logic reduces expressiveness |
| UPPAAL ↔ SystemVerilog | 92 | Synchronization mismatches |
| UPPAAL ↔ DSML | 94 | High fidelity via template-based mapping |
| DSML ↔ C | 91 | Challenges in control-flow recovery |
| DSML ↔ SystemVerilog | 95 | Good mapping of architectural constructs |
| DSML ↔ UPPAAL | 96 | Well-preserved state transitions |

**Table 11**. Edge-case handling success rates.

| Construct | C (%) | SystemVerilog (%) | UPPAAL (%) | DSML (%) |
|---|---|---|---|---|
| Switch Statement | 100 | 100 | 100 | 100 |
| Loops | 85 | 88 | 80 | 90 |
| Mixed Data Types | 88 | 90 | 85 | 95 |

**Table 12**. Memory usage during transformations.

| Source → Target | Memory Usage (MB) |
|---|---|
| C → UPPAAL | 50 |
| C → SystemVerilog | 40 |
| C → DSML | 45 |
| SystemVerilog → C | 42 |
| SystemVerilog → UPPAAL | 48 |
| SystemVerilog → DSML | 35 |
| UPPAAL → C | 55 |
| UPPAAL → SystemVerilog | 50 |
| UPPAAL → DSML | 40 |
| DSML → C | 45 |
| DSML → SystemVerilog | 30 |
| DSML → UPPAAL | 35 |

structures. Transformations involving UPPAAL require additional memory to maintain clocks, synchronization, and state-space information, while DSML transformations are most memory-efficient due to template reuse and high-level abstraction. Overall, memory usage remains **modest (<60 MB)**, confirming the framework's suitability for real-time embedded applications.

**5. Scalability** Transformation performance trends relative to input size and model complexity. The results demonstrate that all transformation pathways scale predictably, with UPPAAL transformations being the most computationally demanding due to timing and synchronization semantics. DSML's abstraction provides superior scalability for complex designs.

- **C-based Transformations**: Latency grows linearly (e.g., 120 ms for 100 LOC to 1.2 s for 1000 LOC).
- **SystemVerilog-based Transformations**: Linear increase, maintaining efficiency even at scale.
- **UPPAAL-based Transformations**: Near-linear growth; scalability influenced by state-space complexity.
- **DSML-based Transformations**: Sublinear growth due to high-level abstractions and template reuse.

The expanded analysis of all transformation routes confirms that the proposed framework maintains reliable and high-quality transformations across heterogeneous notations. Among all directions, transformations involving UPPAAL (both to and from) proved to be the most challenging. These paths required careful management of timed behaviors, synchronization semantics, and the flattening of complex state-based constructs, particularly when transitioning to or from procedural or structural representations like C and SystemVerilog. Nonetheless, the framework maintains good accuracy even in these complex cases.

C transformations show robust handling of logic but need enhancements for complex data structures. SystemVerilog transformations benefit from strong structural mappings and type clarity. DSML transformations, while abstract, provide scalability and ease of high-level modeling. The consistent accuracy, low latency, and broad edge-case handling capabilities make this framework a suitable choice for formal verification, model-based design, and multi-domain software synthesis in embedded system engineering.

Minor information loss occurs due to language mismatches (`extern`, `enum` conversions), but most transformations maintain **semantic integrity**. The **adoption of execution strategies** significantly enhances performance, making the

framework suitable for large-scale embedded system projects. The performance evaluation shows that the framework achieves **high transformation accuracy ($\geq$ 98%)** with minimal information loss, primarily occurring due to **construct mismatches across languages**. Additionally, **low transformation latency ($\leq$ 60ms)** ensures real-time usability, making the framework practical for large-scale embedded system applications.

### 6.2 Complexity analysis of the transformation engine

We analyzed the Transformation Engine in terms of time and space complexity under realistic assumptions about the grammar and transformation workflow. Parsing is performed by ANTLR-generated LL($*$) parsers, and transformation logic is implemented via a single-pass Visitor traversal of the Abstract Syntax Tree (AST). Under well-formed, non-pathological grammars and without excessive use of semantic predicates or backtracking, ANTLR's adaptive LL($*$) parsing behaves linearly with respect to the input size. Consequently, a complete forward or reverse transformation that visits each AST node once has an expected time complexity of $\mathcal{O}(n)$, where $n$ denotes the number of syntactic constructs or tokens processed.

Space complexity is dominated by the AST and the in-memory DSML meta-model; these structures scale approximately linearly with input size, yielding $\mathcal{O}(n)$ space demand. Empirical measurements in Sect 6 corroborate this theoretical assessment. Typical transformations on large lines of embedded C/SystemVerilog required modest memory and completed with low latency. To preserve near-linear behavior in larger or more complex inputs, we recommend (and have implemented where appropriate) incremental parsing, symbol-table-based resolution, and avoidance of expensive global scans during routine transformations.

Memory profiling reveals **efficient resource utilization**, with Java's garbage collection effectively managing memory overhead. Multi-threaded execution improves **scalability**, achieving **1.6× speedup**, though dependencies between constructs necessitate careful execution strategies. Furthermore, the framework robustly handles **edge cases**, such as deeply nested control structures and recursive function calls, ensuring **correctness and stability** across transformations.

While the current implementation demonstrates **strong** empirical validation, future improvements, such as incremental parsing, metadata-based type preservation, and adaptive hybrid execution, can further enhance performance. These refinements will help mitigate minor information loss, optimize processing time, and improve transformation fidelity. In conclusion, the proposed framework provides a **reliable, scalable, and efficient solution** for automated **bidirectional transformations** in embedded systems design and verification, significantly reducing manual effort and improving system consistency across multiple representations.

## 7 Discussion

This study introduces a **blended modeling framework** that enables bidirectional, real-time synchronization across heterogeneous notations, including C, SystemVerilog, Timed Automata (UPPAAL), and DSML. The framework has been evaluated on two industrially relevant case studies, including ventilator control and adaptive cruise control, demonstrating practical applicability and low-latency transformations.

In order to conduct a rigorous and meaningful comparative analysis, several key parameters were identified to evaluate the capabilities, scope, and maturity of existing blended modeling frameworks. These parameters include: (1) **Notation Coverage**, indicating whether the framework supports single, dual, or multiple notations; (2) **Notation Name**, referring to the specific languages or modeling formalisms integrated within the framework; (3) **Blended Modeling**, denoting the extent to which textual and graphical representations are integrated into a unified environment; (4) **Bi-Directional** and **Round-Trip Transformation** support, assessing whether the framework maintains semantic synchronization between heterogeneous representations during iterative design–verification cycles; (5) **Formal Verification**, reflecting the framework's ability to integrate or support formal analysis tools; (6) **Tool Support**, representing the degree of implementation maturity and availability of prototype or industrial-grade tooling; and (7) **Public Availability**, signifying the accessibility

of the framework for replication or community adoption. Based on these parameters, a detailed evaluation of prominent state-of-the-art blended modeling frameworks was conducted, as summarized in **Table 13**.

As shown in Table 13, most existing frameworks offer dual-notation integration, commonly between graphical and textual UML-based representations, enabling partial synchronization but lacking complete round-trip and real-time consistency. Works such as Maro et al. (2015) [24] and Addazi & Ciccozzi (2021) [25] demonstrate blended modeling but are limited to UML notations and partial bidirectionality. Latifaj et al. (2023)[19] extend synchronization toward timed automata and formal verification, though without runtime or low-level language integration. Scheidgen (2008) [27] and Atkinson et al. (2016) [31] depend on on-demand or AST-level synchronization, which may cause semantic drift and delayed updates. The EAST-ADL blended modeling framework (2023) [30], while domain-relevant, lacks verification integration and practical tool support. In contrast, the proposed blended modeling framework (2025) introduces a multi-notation, parser-based approach unifying C, SystemVerilog, UPPAAL, and DSML under a centralized abstract syntax. It supports runtime bidirectional and round-trip transformations while ensuring semantic consistency and formal verification through an integrated, publicly available tool environment. This comprehensive integration bridges the long-standing gap between design, verification, and implementation workflows, providing a scalable, verifiable, and industrially applicable blended modeling solution.

A key challenge in blended modeling is the resolution of semantic mismatches between heterogeneous languages. For example, SystemVerilog timing constructs often need to be approximated when translated into UPPAAL clocks, while preprocessor directives in C have no direct counterparts in SystemVerilog or UPPAAL, requiring encoding as constants or annotations. Type systems further complicate transformations, as constructs such as `uint8\_t` or enumerations are often generalized to integer types, potentially leading to irreversibility in round-trip conversions. Similarly, control-flow differences necessitate restructuring function calls in C into transitions in UPPAAL, introducing further abstraction. To address these challenges, the framework employs approximation strategies, traceability annotations, and hybrid semantic preservation mechanisms, balancing the need for semantic fidelity with practical tractability across heterogeneous languages.

**Table 13**. Comparative analysis of existing blended modeling frameworks.

| Framework/Year | Notation Coverage | Notation Name | Blended Modeling | Bi-Directional | Round-trip | Formal Verification | Tool Support | Public Availability |
|---|---|---|---|---|---|---|---|---|
| Maro et al. (2015) [24] | Dual | UML (Graphical + Textual) | Yes | Yes (Limited) | No | No | Yes | No |
| Addazi & Ciccozzi (2021) [25] | Dual | UML + UML Profiles | Yes | Yes (Limited) | No | No | Prototype | Yes |
| Lazar (2011) [26] | Dual | fUML + Alf | Yes | Yes (Limited) | No | No | Yes | No |
| Scheidgen (2008) [27] | Dual | Embedded Textual + Graphical | Partial | No | No | No | Yes | No |
| Latifaj et al. (2023) [19] | Dual | UML + Timed Automata | Yes | No | No | Yes | Prototype | Yes |
| Atkinson et al. (2016) [31] | Dual | AST-level Synchronization | Partial | Partial | Partial | No | Prototype | No |
| Anwar et al. (2023) [30] | Dual | EAST-ADL + Xtext | Yes | Yes | No | No | Prototype | Yes |
| **Proposed Framework (2025)** | Multi | C, SystemVerilog, UPPAAL, DSML | Yes | Yes | Yes | Yes | Yes | Yes |

The current evaluation demonstrates satisfactory performance in terms of latency, memory usage, and transformation accuracy. Nevertheless, broader industrial validation is required to fully demonstrate the framework's utility. Controlled studies involving multiple development teams in real-world settings will provide quantitative insights into productivity improvements, defect detection, and scalability, thereby complementing the technical metrics presented here.

### 7.1 Limitations and threats to validity

Despite its contributions, the proposed framework has several limitations and potential threats to validity that inform directions for future refinement. The current implementation does not yet support **explicit concurrency modeling**, as constructs such as threads, tasks, and interrupts were deliberately excluded from the selected language subsets. This constrains its applicability in domains where parallelism and real-time responsiveness are central, such as safety-critical embedded systems. Furthermore, **scalability remains an open challenge**. Although correctness and feasibility have been demonstrated through case studies, systematic validation on large-scale industrial codebases and complex model instances is yet to be undertaken. Similarly, the framework currently provides only partial support for **heterogeneous hardware–software co-design**, as maintaining semantic equivalence across behavioral models, hardware description languages, and formal verification tools introduces non-trivial synchronization complexity.

Additionally, while the framework supports **round-trip transformations** among C, SystemVerilog, Timed Automata (UPPAAL), and DSML, **complete integration with external environments**, such as compilers, simulators, and verification tools, remains limited. Users must manually import generated artifacts into corresponding environments (e.g., UPPAAL, GCC, or Simulink), constraining full automation. Minor **information losses** are also inherent due to abstraction differences, execution semantics, and expressiveness mismatches across languages. Constructs such as recursion, polymorphism, inter-process communication, and complex temporal dependencies are not yet comprehensively handled, which may affect transformation fidelity in specialized contexts.

From a validity perspective, several risks were identified. **Construct validity** may be affected because certain timing-sensitive behaviors in UPPAAL cannot be perfectly mirrored in untimed languages, possibly reducing representational fidelity. **Internal validity** may be influenced by parser limitations in handling rare constructs or dynamic behaviors. **External validity** is constrained by the domain scope of current case studies, which may not generalize to aerospace or robotics systems without rule adaptation. **Conclusion validity** could also be impacted by environmental factors such as JVM configurations or tool dependencies that influence runtime behavior.

These limitations and validity concerns collectively emphasize the need for cautious interpretation of results while providing a roadmap for future work focused on improving scalability, semantic precision, concurrency modeling, and seamless toolchain integration.

### 7.2 Future work

Building upon the current foundation, future work will address these limitations and enhance both the expressiveness and industrial relevance of the framework. One important direction is the **integration of concurrency constructs**. Threads, tasks, and interrupts will be modeled explicitly, and synchronization channels will be enhanced in UPPAAL to capture parallel execution semantics accurately. Alongside these modeling improvements, advanced verification techniques, including compositional reasoning and assume–guarantee analysis, will be incorporated to manage the increased state space complexity introduced by parallel constructs.

Another focus will be **large-scale industrial validation**. Planned studies with multiple partners in automotive, aerospace, and medical domains will examine the framework's scalability, productivity benefits, and defect detection

capabilities in real-world projects. These evaluations are intended to provide empirical evidence for the framework's robustness and to guide refinements in transformation strategies, tool integration, and user interaction design.

Further enhancements will explore **hybrid approaches for heterogeneous hardware–software co-design**, enabling synchronized transformations across behavioral models, hardware descriptions, and formal verification artifacts. This will extend the framework's applicability to domains with tightly coupled hardware–software interactions, which are prevalent in modern embedded systems.

Finally, the framework will be **extended to support richer language subsets and improved usability**. Additional notations, including VHDL, MATLAB Simulink, and AADL, will be incorporated to accommodate diverse industrial modeling practices. At the same time, the GUI will evolve into a fully interactive environment, providing visualization, live simulation feedback, and seamless integration with external verification and code-generation tools. These enhancements aim to reduce manual intervention, improve semantic fidelity, and facilitate adoption in industrial engineering workflows.

Collectively, these future directions will strengthen the framework's capability to support bidirectional transformations, ensure semantic fidelity across heterogeneous notations, and provide a scalable, usable tool for complex, safety-critical embedded system development.

### 7.3 Tool support and reproducibility

The current framework provides round-trip transformation support across C, SystemVerilog, Timed Automata (UPPAAL), and DSML tree notations. However, full external tool collaboration (e.g., direct simulation in UPPAAL, compilation in GCC, or co-simulation with Simulink) is not automated. Users are required to manually import the generated artifacts into their respective environments. For compatibility, the framework has been tested and validated with **UPPAAL 4.1.26**, **GCC 11.2**, **SystemVerilog IEEE 1800-2017**, and **ANTLR 4.13.0**.

Conclusively, enhancing transformation fidelity remains a key priority, especially for semantics-rich languages like UPPAAL. Future improvements will aim to support complex constructs such as recursion, polymorphic types, inter-process communication, and temporal behaviours. Techniques like refined parsing strategies, timing abstraction, and consistency rule checking will be incorporated to minimize semantic loss and manual intervention during transformations. From a usability standpoint, the current GUI prototype will evolve into a comprehensive modeling environment. This includes visualization features, live simulation feedback, and seamless integration with external verification tools. These improvements will support broader industrial adoption by making the framework more interactive, traceable, and aligned with existing model-driven engineering toolchains.

### 8 Conclusion

This research proposes a robust and extensible bidirectional transformation framework for synchronized modeling across multiple notations in embedded system design and verification. It addresses the critical challenge of semantic consistency among heterogeneous languages by supporting round-trip transformations, ensuring the fidelity of structure and logic across models. Minor, noncritical information loss (e.g., omission of certain keywords) was observed, yet the framework preserved core semantics effectively.

Distinct from unidirectional or UML-reliant methods, this implementation-level approach caters to the real-time demands of embedded systems by accommodating low-level constructs and domain-specific notations. The efficiency of the framework was validated through empirical results. Validity threats were systematically addressed through construct, internal, and external evaluations using diverse case studies such as ventilator and cruise control systems. These assessments demonstrate the robustness and adaptability of the framework across contexts. The framework thus contributes to a scalable and practical foundation for the development of next-generation embedded systems.

## 9 Appendix A

### 9.1 Grammar definition

Defining a formal grammar is fundamental for accurately parsing and transforming programming and modeling languages. Given the subsets identified in the previous section, a concrete syntax grammar must capture the structural semantics of C, SystemVerilog, Timed Automata, and DSML while ensuring syntactic correctness and transformation feasibility. Various grammar-based mechanisms exist, including context-free grammars (CFGs), attribute grammars, and parsing expression grammars (PEGs). Context-free grammars (CFGs) are traditionally used to define the syntax of programming languages, while attribute grammars extend CFGs by associating semantic rules with syntax productions. Parsing Expression Grammars (PEGs) offer deterministic parsing, although they may introduce additional complexity when dealing with ambiguity.

ANTLR (Another Tool for Language Recognition) has been adopted as the primary grammar specification tool in this research. Its support for LL(*) parsing, modular grammar construction, and automated parse tree generation makes it suitable for handling multiple language syntaxes within a unified framework. This work introduces a novel ANTLR-based grammar to support the parsing and transformation of C, SystemVerilog, Timed Automata (UPPAAL), and DSML Meta-Model representations. This grammar forms the backbone of the automated transformation engine, facilitating formal verification and model-driven development.

Several factors contribute to the selection of ANTLR in this context:

- **Precise Syntax Definition:** Enables structured and syntactically accurate grammar specifications for each target language.
- **Modular and Extensible Design:** Supports easy adaptation and scalability to additional languages or constructs without requiring major redesigns.
- **Error Handling Capabilities:** Built-in mechanisms provide structured reporting and facilitate debugging.
- **Cross-Domain Applicability:** Integrates software, hardware, and formal modeling languages under a single transformation framework.

ANTLR operates in three primary phases. The first phase, **Lexical Analysis (Tokenization)**, involves the lexer (scanner) breaking the input source code into tokens such as keywords, identifiers, and operators. Each token is then assigned a specific type, such as `INT`, `IF`, or `IDENTIFIER`. The second phase, **Parsing (Syntax Analysis)**, applies grammar rules to recognize syntactic patterns within the token sequences. During this process, the parser constructs a **parse tree**, ensuring that the input adheres to the predefined grammar rules. The final phase, **Abstract Syntax Tree (AST) Generation**, involves traversing the parse tree using the **visitor** or **listener** pattern to create an AST. This structured representation of the code is later utilized for transformations, enabling efficient processing and conversion into the desired target format. ANTLR components are summarized in Table 14, for reference.

ANTLR grammars were created for each language subset (C language, SystemVerilog, Timed Automata, and Meta-Model DSML) to provide a robust mechanism for parsing and validating input code.

**Table 14**. ANTLR components.

| ANTLR Component | Description |
|---|---|
| Lexer | Tokenizes the input source code into **meaningful symbols**. |
| Parser | Uses grammar rules to generate a **parse tree**. |
| Listener | Uses **event-driven traversal** of the parse tree (auto-generated by ANTLR). |
| Visitor | Uses **custom tree traversal logic** for transformations. |
| AST (Abstract Syntax Tree) | Represents the **hierarchical structure** of the source code, aiding transformation. |

- **C Grammar**: Defines the syntax and semantics of C constructs, including declarations, loops, and function calls.
- **SystemVerilog Grammar**: Captures the syntactical rules for hardware modeling, verification logic, and assertion constructs.
- **Timed Automata Grammar**: Specifies the structure for modeling timed state transitions, clocks, and timing constraints.
- **DSML Grammar**: Focuses on high-level abstractions and visual representations tailored to embedded systems.

Each grammar was carefully designed to enable bidirectional transformations while minimizing ambiguities, maximizing consistencies, and ensuring compatibility with the domain model. To maintain conciseness and avoid redundancy, this paper presents only the definition of grammar, transformation rules, and illustrative examples of the **C language**. Due to space limitations, the complete grammar specifications and transformation rules for the remaining notations (SystemVerilog, Timed Automata -UPPAAL, and DSML) are not included herein. However, these resources are fully documented and publicly available at the provided GitHub repository, which can be accessed by following reference [35].

**C Language grammar.** The C grammar is designed to parse a subset of C language constructs (as identified earlier) and facilitate their transformation into other notations through abstract and concrete syntaxes. It is structured into three key components: lexical analysis (tokenization), parsing (syntax analysis), and AST generation. These align with ANTLR's standard processing phases. The rules below follow a BNF-style representation with corresponding descriptions.

1. **Lexical Analysis (Tokenization)** The lexer defines rules for recognizing tokens:

- **Keywords**: `IF`, `FOR`, `SWITCH`, `RETURN`, `VOID`, `INT`, `FLOAT`, etc.
- **Identifiers**: `<ID>`
- **Constants**: `<INT>`, `<HEX_INT>`
- **Operators**: `+` (`PLUS`), `-` (`MINUS`), `==` (`EQUAL`), `&&` (`AND`), `||` (`OR`)
- **Delimiters**: `;` (`SEMICOLON`), `,` (`COMMA`), `{}`, `[]`, `()`
- **Comments**: `//` (`SINGLE_LINE_COMMENT`), `/* */` (`MULTI_LINE_COMMENT`)

Tokens are the basic building blocks of C source code and are essential for breaking down the code into meaningful components before syntax parsing.

2. **Parsing Rules (Syntax Analysis)** The grammar rules below describe a subset of C used for transformation into other notations. The rules are written in BNF style with corresponding natural language explanations.

**Rule 1.** `<File> ::=<PreprocessorDirective> <IncludeDirective> <ModuleDecl>`

This rule defines the top-level structure of the file. A file can contain preprocessor directives, include directives, or module declarations.

**Rule 2.** `<PreprocessorDirective> ::= #define <ID> [= <INT>] [;]`

A preprocessor directive can be a define statement with an optional assignment and a semicolon.

**Rule 3.** `<IncludeDirective> ::= include <SystemInclude> | <LocalInclude>`

Include directives allow the inclusion of system headers or local headers in angle or double quotes.

**Rule 4.** `<ModuleDecl> ::= <ModuleItem>`

A module declaration consists of a single module item, such as a declaration, function, or statement.

**Rule 5.** `<ModuleItem> ::= <Declaration> | <FunctionDecl> | <Statement> | <EnumDecl>`

A module item can be a variable declaration, function declaration, statement, or enumeration.

**Rule 6.** `<FunctionDecl> ::= <ReturnType> <ID> ( [<ParameterList>] )[;] [{<Statement>* }]`

This rule defines a function with return type, name, optional parameters, and an optional body of statements.

**Rule 7.** `<ReturnType> ::= int | uint8_t | uint32_t | osThreadId | char | TickType_t | osEvent | float | void | <ID>`

A return type can be one of the common types or a user-defined identifier.

**Rule 8.** `<ParameterList> ::= * [* <ID>] [ , * ]`

A parameter list can have one or more parameters, optionally followed by a pointer identifier.

**Rule 9.** ` ::= <ID> | void | const | <DataType>`

Each parameter can be an identifier, 'void', 'const', or a data type.

**Rule 10.** `<Declaration> ::= [extern] [const] <DataType> <ID> [ [ <INT> ] ] [= <Primary>] ;`

A variable declaration may be preceded by extern or const, followed by a type, name, optional array size, and an optional initializer.

**Rule 11.** `<DataType> ::= int | uint8_t | uint32_t | osThreadId | char | TickType_t | osEvent | float | <ID>`

A data type can be a primitive type or a user-defined identifier.

**Rule 12.** `<EnumDecl> ::= typedef enum { <EnumList> } <ID> ;`

An enumeration is declared using the typedef keyword followed by a list of identifiers and the enum name.

**Rule 13.** `<EnumList> ::= <ID> ( , <ID> )*`

A list of comma-separated identifiers inside an enum.

**Rule 14.** `<Statement> ::= <Assignment> | <Declaration> | <FunctionCall> | <IfStatement> | <LoopStatement> | <Switch Statement> | <PrintStatement> | <Comment> | <Return>`

This rule defines the different types of executable statements in the language.

**Rule 15.** `<ReturnStatement> ::= return [<Expression>] ;`

A return statement optionally returns a value.

**Rule 16.** `<Assignment> ::= <IndexedID> = <Expression> | <AssignmentType> <ID | INT>`

An assignment modifies a variable or array element, optionally using compound assignment operators.

**Rule 17.** `<AssignmentType> ::= ++ | -- | -= | += | =- | =+`

Compound assignment types like increment, decrement, and arithmetic updates.

**Rule 18.** `<IndexedID> ::= <ID> [ [ <Expression> ] ]*`

An indexed identifier can be a simple variable or an array access.

**Rule 19.** `<FunctionCall> ::= <ID> ( [<Argument List>] )`

A function call contains a function name and an optional list of arguments.

**Rule 20.** `<ArgumentList> ::= <Argument> ( , <Argument> )*`

A list of comma-separated arguments.

**Rule 21.** `<Argument> ::= <Expression> | <TransformedArgument>`

An argument can be an expression or a special transformed form.

**Rule 22.** `<TransformedArgument> ::= <DataCast Argument> | <FunctionCall Argument>`

Transformed arguments can involve casting or nested function calls.

**Rule 23.** `<DataCastArgument> ::= ( <DataType> * )<ID>`

A data type cast expression.

**Rule 24.** `<FunctionCallArgument> ::= <ID> ( <ID> )`

A function call is used as an argument.

**Rule 25.** `<IfStatement> ::= if ( <Expression> ){ <Statement>* } [else { <Statement>* } | else <IfStatement>]`

An if statement can optionally include an else clause with either a block or a nested if.

**Rule 26.** `<LoopStatement> ::= <ForLoop> | <WhileLoop>`

Loop statements can be either for-loops or while-loops.

**Rule 27.** `<ForLoop> ::= for ( [volatile] <DataType>`

A for loop includes initialization, condition, and increment expressions inside parentheses.

**Rule 28.** `<WhileLoop> ::= while ( <Expression> ){ <Statement>+ }`

A while loop evaluates a condition and executes a block of statements repeatedly.

**Rule 29.** `<SwitchStatement> ::= switch ( <Expression> ){ <CaseBlock>+ [<DefaultBlock>] }`

A switch statement uses cases and an optional default block to handle multi-branch logic.

**Rule 30.** `<CaseBlock> ::= case <Expression> : { <Statement>* break ; }`

Each case block handles a specific value and ends with a break.

**Rule 31.** `<DefaultBlock> ::= default : <Statement>* break ;`

A default block handles cases not matched by any specific case.

**Rule 32.** `<PrintStatement> ::= printf ( [<STRING>] [, <Argument>] [, <ArgumentList>] )`

A print statement using printf syntax can include a string and multiple arguments.

**Rule 33.** `<Comment> ::= // <text> | /* <text> */`

Single-line or multi-line comments.

**Rule 34.** `<Expression> ::= [&]? <Primary> ( ( <ArgumentList> ))? ( <Operator> <Primary> )*`

An expression can optionally be a reference, involve function calls, and use binary operators.

**Rule 35.** `<Primary> ::= <ID> | <HEX_INT> | <INT> | <STRING> | <CHAR> | <CastFunction Call>` `| <DataType> | ( <Expression> )`

A primary element of an expression can be an identifier, a literal, or a nested expression.

**Rule 36.** `<Operator> ::= + | - | * | / | % | < | > | <= | >= | == | = | &&!`

Operators for arithmetic, relational, equality, and logical operations.

**Rule 37.** `<CastFunctionCall> ::= ( <DataType> )<ID> ( )`

A cast function call consists of a data type cast followed by a function call.

**Rule 38.** `<PointerDeclaration> ::= <DataType> * <ID> [= <Expression>] ;`

A pointer declaration consists of a data type followed by an asterisk to denote the pointer, an identifier as the variable name, and an optional initialization using an expression.

This rule-based format allows clear mapping between grammar components and transformation logic in C-to-other notation conversions.

3. **Abstract Syntax Tree (AST) Generation Example 1: Function Declaration and Compound Statement**

```
void SetMode(int mode)
{
   Mode = mode;
   osDelay(100);
}
```

**Applied Rules:**

- **Rule 7**: `<FunctionDecl> ::= <ReturnType> <ID> ( <ParameterList>? )[;] ({ < Statement>* })?`
- **Rule 8**: `<ReturnType> ::= void`
- **Rule 9/10**: Parameter list with `<DataType>` and `<ID>` → `int mode`
- **Rule 14/39**: `<CompoundStatement>` contains two `<Statement>`s inside `{}`
- **Rule 12**: `<Statement>` for assignment
  (`Mode = mode;`) and function call (`osDelay(100);`)

**Explanation:** This example defines a function `SetMode` with a `void` return type and one integer parameter. Its body is a compound block containing two statements: an assignment and a delay function call. The parse tree for this structure would have a root node for `FunctionDecl`, branching into `ReturnType`, `ID`, `ParameterList`, and `CompoundStatement` as shown in **Fig 10**. In the figure, the **orange boxes** connected via **green arrows** represent the

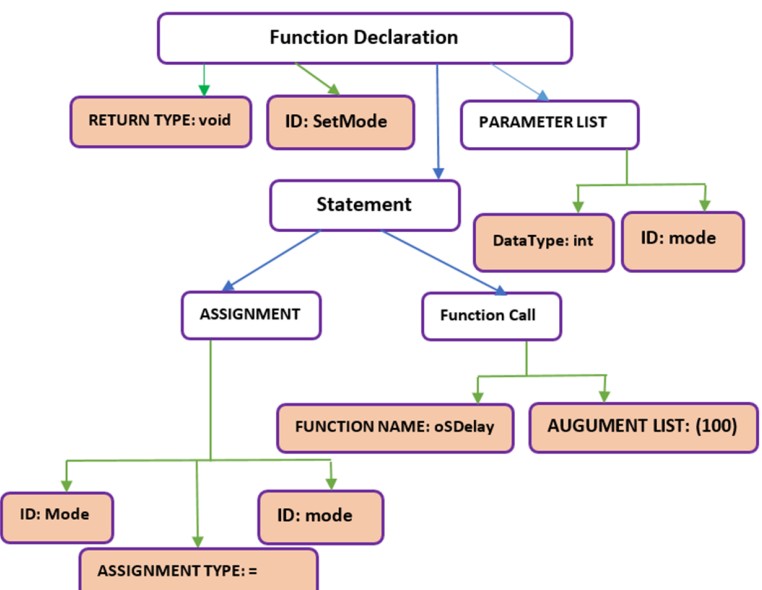

**Fig 10**. Abstract Syntax Tree (AST) representation for Example 1.

**leaf nodes** of the AST. These are terminal symbols from the grammar, such as `void`, `SetMode`, `int`, `mode`, `Mode`, `=`, and `100`. They are **lexical elements** identified after parsing, with no further syntactic breakdown. The **purple-bordered white boxes** represent non-terminal nodes that group related structures, mapping directly to grammar rules (like `<FunctionDecl>`, `<Assignment>`, `<FunctionCall>`).

**Example 2: Switch Statement and Enum**

```
typedef enum {IDLE, RUNNING, ALARM} State;
switch (state)
{ case IDLE:
      prepare(); break;
  case RUNNING:
      monitor(); break;
  case ALARM:
      alert(); break;
}
```

**Applied Rules:**

- **Rule 6**: `<EnumDecl> ::= typedef enum { <EnumList> } <ID>;`
- **Rule 12**: `<Statement>` → SwitchStatement
- **Rule 20**: `<FunctionCall>`s inside cases

**Explanation:** This example shows a user-defined `enum` type named `State`, and a switch-case control structure operating on a variable `state`. Each case invokes a corresponding function. This composite structure demonstrates rule composition with nested function calls inside branching logic, which corresponds to subtrees in the AST, much

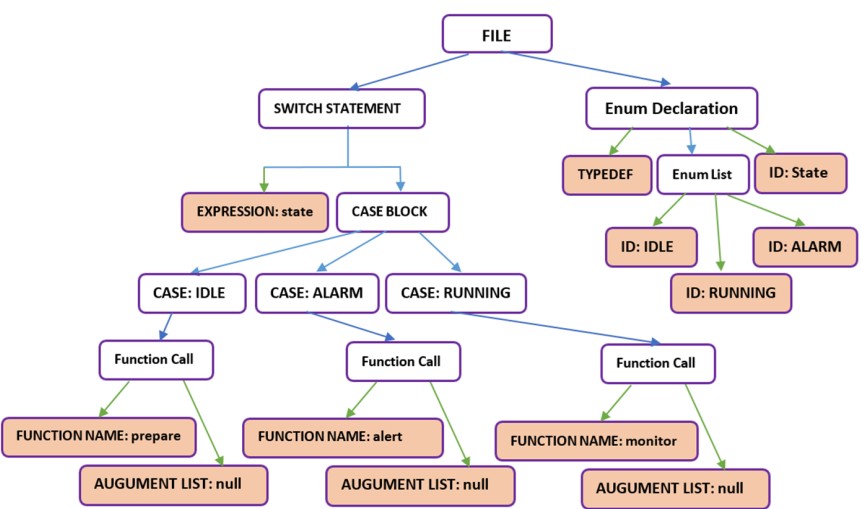

**Fig 11**. **Abstract Syntax Tree (AST) representation for Example 2.**

like those depicted in Fig 11. The Abstract Syntax Tree (AST) for Example 2 represents both the enum declaration and the switch-case control logic. At the top level, the AST begins with a `FILE` node, branching into two main components: `Enum Declaration` and `SWITCH STATEMENT`. The `Enum Declaration` node further expands into `TYPEDEF`, an `Enum List`, and individual `ID` nodes representing the enumerated values: `IDLE`, `RUNNING`, and `ALARM`, all grouped under the `State` type. On the other side, the `SWITCH STATEMENT` node is composed of an `EXPRESSION` node (with the variable `state`) and a `CASE BLOCK`. Each `CASE` node, `IDLE`, `RUNNING`, and `ALARM`, is associated with a `Function Call` node. These function calls (`prepare()`, `monitor()`, and `alert()`) each have a `FUNCTION NAME` and an `ARGUMENT LIST`, which is `null` in all cases here. The AST structure clearly captures how the enum values control the logic flow through a switch-case mechanism, invoking different functions based on the current `state`.

This **C grammar** effectively captures C syntax and enables seamless transformation into other representations through a structured grammar parsing pipeline.

## Author contributions

**Conceptualization:** Misbah Mehboob Awan, Muhammad Waseem Anwar, Wasi Haider Butt.

**Formal analysis:** Muhammad Waseem Anwar, Wasi Haider Butt, Farooque Azam.

**Funding acquisition:** Farooque Azam.

**Investigation:** Misbah Mehboob Awan.

**Methodology:** Misbah Mehboob Awan, Muhammad Waseem Anwar.

**Project administration:** Wasi Haider Butt, Farooque Azam.

**Resources:** Misbah Mehboob Awan.

**Software:** Misbah Mehboob Awan.

**Supervision:** Muhammad Waseem Anwar, Wasi Haider Butt.

**Validation:** Misbah Mehboob Awan, Wasi Haider Butt, Farooque Azam.

**Writing – original draft:** Misbah Mehboob Awan.

**Writing – review & editing:** Misbah Mehboob Awan, Muhammad Waseem Anwar, Wasi Haider Butt, Farooque Azam.

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
