## [Decision Letter · Decision Letter 0]

15 Sep 2025

PONE-D-25-39470A blended modeling framework for real-time design and verification of safety-critical embedded systemsPLOS ONE

Dear Dr. Mehboob Awan,

Thank you for submitting your manuscript to PLOS ONE. After careful consideration, we feel that it has merit but does not fully meet PLOS ONE’s publication criteria as it currently stands. Therefore, we invite you to submit a revised version of the manuscript that addresses the points raised during the review process.

We look forward to receiving your revised manuscript.

Kind regards,

Asadullah Shaikh, Ph.D.

Academic Editor

PLOS ONE

Journal Requirements:

“This work is partially supported by the Higher Education Commission, Pakistan, through the NRPU MRED project under Grant No. [20-15651].”

“This work is partially supported by the Higher Education Commission, Pakistan, through the NRPU MRED project under Grant No. [20-15651].”

“This work is partially supported by the Higher Education Commission, Pakistan, through the NRPU MRED project under Grant No. [20-15651].”

“This work is partially supported by the Higher Education Commission, Pakistan, through the NRPU MRED project under Grant No. [20-15651].”

Reviewers' comments:

Reviewer's Responses to Questions

**Comments to the Author**

1. Is the manuscript technically sound, and do the data support the conclusions?

Reviewer #1: Yes

Reviewer #2: Yes

Reviewer #3: Yes

Reviewer #4: Yes

2. Has the statistical analysis been performed appropriately and rigorously?

Reviewer #1: N/A

Reviewer #2: No

Reviewer #3: No

Reviewer #4: Yes

3. Have the authors made all data underlying the findings in their manuscript fully available?

Reviewer #1: Yes

Reviewer #2: Yes

Reviewer #3: Yes

Reviewer #4: Yes

4. Is the manuscript presented in an intelligible fashion and written in standard English?

Reviewer #1: Yes

Reviewer #2: Yes

Reviewer #3: Yes

Reviewer #4: Yes

5. Review Comments to the Author

Reviewer #1: Reviewer Comments

The manuscript presents an ambitious and relevant contribution by proposing a blended modelling framework for real-time design and verification of safety-critical embedded systems. The integration of C, System Verilog, Timed Automata, and DSML through bidirectional transformations is timely and impactful, particularly for industrial contexts such as ventilator and cruise control systems. However, while the framework shows promise, several aspects of the manuscript would benefit from deeper clarification, stronger validation, and tighter presentation. The following comments are intended to help the authors strengthen the technical rigor and readability of the paper before resubmission.

1. The novelty of the work is promising, but the distinction from existing blended modeling frameworks such as HOT-based synchronization, EMF-based solutions, and AUTOSAR workflows is not emphasized strongly enough. A comparative table with technical differentiators would clarify the contribution.

2. The architecture description is detailed but too descriptive; a layered workflow diagram showing data exchange across C, SystemVerilog, Timed Automata, and DSML would make the framework clearer. Complexity analysis of the transformation engine (time and space cost) should also be added.

3. The rationale for language subset selection is qualitative. Empirical justification such as coverage statistics of industrial code or representative benchmarks is required. Excluding concurrency constructs like threads, tasks, and interrupts limits applicability for safety-critical systems; this requires justification and discussion of future extension.

4. Transformation rules need deeper discussion of semantic equivalence issues, e.g., mapping System Verilog timing constructs to Timed Automata clocks. Examples of constructs where exact mapping was not possible, and the strategies used to approximate them, should be presented. Quantitative data on information loss during reverse transformations would strengthen the claims.

5. Case studies are appropriate, but evaluation metrics are too narrow. In addition to latency, memory, and transformation accuracy, please include productivity gains (developer time saved), defects detected or prevented due to synchronization, and scalability results (lines of code or model size handled). Comparisons with baseline workflows or existing frameworks would highlight practical improvements.

6. Tool support and reproducibility need more clarity. Specify whether the tool supports round-trip editing in all notations, what dependencies (ANTLR version, UPPAAL integration) are required, and provide screenshots or GUI demonstration figures.

7. The manuscript is overly long. Detailed grammar rules and AST breakdowns could be moved to supplementary material, while keeping only key illustrative examples in the main text. Figures should have clearer captions explaining their relevance to the framework.

8. Limitations are under-stated. Explicitly discuss current lack of concurrency modeling, scalability to larger industrial systems, and potential challenges in heterogeneous hardware–software co-design. A short future work section would better position the research.

Overall, the paper has strong potential, but revisions are needed to clarify novelty, strengthen validation, and sharpen the technical depth of transformation and semantic fidelity claims.

Reviewer #2: An in-depth analysis of the manuscript reveals a high-quality research work, characterized by clarity of presentation, methodological rigor, technical soundness, and a strong alignment between the data presented and the conclusions reached.

The manuscript is presented clearly and intelligibly, written in standard technical English, consistent with scientific publications in the field of software engineering and embedded systems. The structure follows the conventional academic format, with logical sections that guide the reader from the contextualization of the problem to the presentation of the results and conclusions, facilitating understanding of the work.

One of the strongest points of the work is the authors' commitment to transparency and reproducibility. They have made the complete implementation of the proposed framework fully available to promote future research. The source code, including grammars, transformation rules, "visitor" classes, and the graphical interface, is publicly accessible in a GitHub repository. Additionally, the authors have provided a pre-compiled executable for easy use and verification of the results without the need for compilation. The article does not perform a formal statistical analysis (with hypothesis testing, for example), which is common in other scientific fields. Instead, the authors conduct a rigorous empirical performance evaluation, which is the appropriate methodology for validating the effectiveness of a software framework like this. The evaluation was based on multiple well-defined parameters, such as transformation latency, round-trip transformation accuracy, edge case handling, memory usage, and scalability. The analysis presented is systematic and suitable for measuring the feasibility and efficiency of the proposed solution in the context of embedded systems.

The manuscript is technically sound, based on well-established principles. The solution architecture employs robust, industry-standard tools and techniques, such as:

The use of a central abstract syntax (DSML meta-model) to ensure semantic consistency across different languages.

The use of the ANTLR tool for language parsing, which ensures a robust and reliable code interpretation process. The use of appropriate design patterns, such as the Visitor Pattern, to apply transformation rules in a modular and extensible manner.

The robustness of the work is reinforced by the transparent discussion of the technical challenges inherent to the approach, such as the inevitable loss of information in bidirectional transformations, which is analyzed in detail.

More importantly, the data presented convincingly corroborate the paper's conclusions.

The claims of efficiency and practicality are supported by low latency times (mostly below 130 ms for 100 lines of code) and modest memory usage.

The robustness and adaptability of the framework are demonstrated by successful validation on two industrial case studies from distinct domains—a medical ventilator system and an automotive cruise control system.

The main conclusion that the framework maintains semantic fidelity with minor losses is directly supported by the accuracy data, which shows hit rates above 90% for most transformation paths, while the analysis identifies and explains the exact sources of these minor losses. In conclusion, the combined analysis indicates that the manuscript is an exemplary piece of research. It is well-written, technically sound, transparent about its limitations, and, crucially, its conclusions are strongly supported by the empirical data presented. Despite the framework's robustness, future improvements could focus on mitigating the small information losses that occur during round-trip transformations and expanding the currently supported language subsets to encompass more complex and rare code constructs. Validating the approach in larger-scale and more complex industrial projects would serve to more rigorously test its scalability and robustness in real-world scenarios. Additionally, improvements in the translation of rich temporal semantics (such as those of Timed Automata) would significantly increase its applicability and impact in industry.

Reviewer #3: This research work theme translates between multiple languages useful in embedded design. The paper is well-written and contributions are explained in a logical manner. The following comments may improve the quality of work.

. The artwork needs a revision for visibility. A few of the included pictures are barely readable.

. Inclue a table summarizing related research outlined in section 2.

. Consider defining: Design, verificiation, validation, modeling, etc.

. Abbreviations first appeared in text should be in full term.

. It is not clear from the text whether the author implemented the code translation for the full application code or just a part of it.

. Discussion section is very brief.

. Traces of using GenAI needs to be removed from the paper, "This rule-based format allows clear mapping between grammar components and transformation logic in C-to-other notation conversions. Let me know if you need a matching SystemVerilog target grammar or transformation rules next."

. Another major change which needs to be addressed is to revise the design. It mostly consists of documents and the code is not a primary part of it. Once, the design is finalized; appropriate language (or combination) may be used for application coding. The coding is subjected to verification (testing).

The above comments may improve the quality of research work.

Reviewer #4: This work proposes a blended modeling framework that provides bidirectional automatic conversion between different representations (C, SystemVerilog, Timed Automata, DSML) for the design and verification processes of embedded systems. Its strength lies in its real-time synchronization between the different representations and its demonstration on two industrial examples: a ventilator and cruise control. Furthermore, performance analyses demonstrate that the conversions operate with low latency and reasonable memory consumption.

However, some limitations are noted in the article:

First of all, the study appears to be quite long, which makes it difficult to read. It also gives the impression of being a book chapter rather than an article. Therefore, the authors should address this issue. Perhaps some sections could be included in the Appendix section.

The scope of the case studies appears to be limited and supplementation with larger/complex industrial systems could have strengthened the generalizability of the method.

Furthermore, no comparative evaluation of the framework with existing industry standards (e.g., AUTOSAR, Simulink) has been conducted, making it difficult to clarify the practical advantages of the proposed approach.

Reference 33 cannot be used in an academic study.

The discussion section should also compare the study with the literature and highlight its strengths and weaknesses. This will help transition from this point to the future vision more accurately. It would be even better if a summary table and commentary could be provided here.

The values given in tables 9, 10 and 11 in Section 6 need to be explained along with their reasons.

6. PLOS authors have the option to publish the peer review history of their article (what does this mean?). If published, this will include your full peer review and any attached files.

Reviewer #1: No

Reviewer #2: No

Reviewer #3: No

Reviewer #4: **Yes: **serkan dereli

---

## [Author Response · Author response to Decision Letter 1]

20 Oct 2025

Reviewer 1

R1.0 The manuscript presents an ambitious and relevant contribution by proposing a blended modelling framework for real-time design and verification of safety-critical embedded systems. The integration of C, System Verilog, Timed Automata, and DSML through bidirectional transformations is timely and impactful, particularly for industrial contexts such as ventilator and cruise control systems. However, while the framework shows promise, several aspects of the manuscript would benefit from deeper clarification, stronger validation, and tighter presentation. The following comments are intended to help the authors strengthen the technical rigor and readability of the paper before re- submission.

Author’s response: We sincerely thank the reviewer for their thoughtful and encouraging assessment of our work. We are grateful that the reviewer recognizes the ambition, relevance, and industrial applicability of our proposed blended modeling framework, particularly its ability to integrate C, SystemVerilog, Timed Automata, and DSML through bidirectional transformations for safety-critical contexts such as ventilator and cruise control systems. We also deeply appreciate the constructive feedback regarding the need for deeper clarification, stronger validation, and tighter presentation. These observations are invaluable in guiding us to improve both the technical rigor and the readability of the manuscript. In the revised version, we have carefully addressed each of the reviewer’s detailed comments point by point, strengthening the novelty claims, enhancing the evaluation and validation components, and improving the clarity of figures, tables, and narrative flow. We believe these revisions substantially improve the overall quality and accessibility of the paper.

R1.1 The novelty of the work is promising, but the distinction from existing blended modeling frameworks such as HOT-based synchronization, EMF-based solutions, and AUTOSAR workflows is not emphasized strongly enough. A comparative table with technical differentiators would clarify the contribution.

Author’s response: We sincerely thank the reviewer for this valuable observation. We fully agree that the distinction between our proposed framework and existing blended modeling approaches re- quires clearer articulation. Accordingly, we have substantially revised the Discussion section to include a comparative analysis table (Table 13) that systematically contrasts our framework with existing blended modeling approaches.

The updated table and its accompanying narrative highlight the key differentiators of our research, including the use of a parser-based transformation engine, runtime bi-directional synchronization, and multi- notation coverage across C, SystemVerilog, UPPAAL, and DSML, capabilities not concurrently supported in existing frameworks. This enhancement strengthens the novelty, clarity, and scientific positioning of the proposed work within the broader context of blended modeling research.

R1.2 The architecture description is detailed but too descriptive; a layered workflow diagram showing data exchange across C, SystemVerilog, Timed Automata, and DSML would make the framework clearer. Complexity analysis of the transformation engine (time and space cost) should also be added.

Author’s response: We sincerely thank the reviewer for this insightful suggestion. As correctly noted, the architecture description in its current form is already detailed, and introducing additional layered diagrams would risk overcrowding the figure. Importantly, the bi-directional exchange of information across notations (C, SystemVerilog, Timed Automata, and DSML) is already visually represented in our framework diagram within the Transformation Engine and Multi-Representation Editor components, where synchronization takes place in real-time. The first stage (Language Subset Evaluation and Selection) does not involve data exchange but serves to scope the subset of constructs for transformation, and this has now been explicitly clarified in the text to avoid ambiguity. To further strengthen the manuscript, we have added a more explicit textual explanation of data exchange across notations in Section 3.1 (highlighted paragraph), ensuring clarity without compromising readability. Regarding the re- viewer’s suggestion for complexity analysis, we agree that this is an essential addition. Accordingly, Section 6 (Performance Evaluation) has been revised to include a complexity analysis of the Transformation Engine in terms of both time and space cost. In particular, parsing and transformation are shown to operate in linear time O(n) with respect to the size of the input program, while memory requirements for intermediate structures (AST and DSML meta-model) also scale linearly. Empirical results in Section 6 confirm this theoretical analysis, demonstrating low memory usage and transformation latency, thereby validating the scalability and efficiency of the approach. We believe these clarifications and additions directly address the reviewer’s concern and significantly enhance the technical rigor of the manuscript.

R1.3 The rationale for language subset selection is qualitative. Empirical justification, such as coverage statistics of industrial code or representative benchmarks, is required. Excluding concurrency constructs like threads, tasks, and interrupts limits applicability for safety-critical systems; this requires justification and discussion of future extension.

Author’s response: We sincerely thank the reviewer for this valuable comment. In response, we have extended the manuscript to strengthen the empirical justification of the selected language subsets. Specifically, we have added Section 3.3.2 (Empirical Coverage of Selected Subsets), which provides quantitative analysis of the representative- ness of the chosen constructs based on industrial code benchmarks. To further support this, we have introduced Table 4 (Coverage of Selected Language Subsets in Industrial Benchmarks), which clearly demonstrates that the chosen subset provides broad coverage for typical embedded system designs. Relevant studies have been cited to substantiate this analysis (Refs. 35 and 36). Regarding the exclusion of concurrency constructs such as threads, tasks, and interrupts, we acknowledge their importance in safety-critical contexts. To address this, we have explicitly discussed the rationale for their omission in the current phase, while outlining concrete strategies for their incorporation in subsequent work. A dedicated paragraph has been added in the Future Directions section, highlighting that extending our framework to handle concurrency constructs remains an essential next step toward broader applicability. All of these additions have been highlighted in the revised manuscript for the reviewer’s convenience.

R1.4 Transformation rules need deeper discussion of semantic equivalence issues, e.g., mapping System Verilog timing constructs to Timed Automata clocks. Examples of constructs where exact mapping was not possible, and the strategies used to approximate them, should be presented. Quantitative data on information loss during reverse transformations would strengthen the claims.

Author’s response: We sincerely thank the reviewer for this insightful observation. We would like to clarify that the discussion on semantic equivalence and information loss in transformation rules has already been incorporated in the manuscript under Section 3.5: Critical Analysis of Round-Trip Transformations in Blended Modeling. However, this section has now been further refined and expanded to explicitly address the reviewer’s concern and enhance clarity. This section explicitly analyzes the inherent semantic mismatches that arise when mapping constructs across C, SystemVerilog, Timed Automata (UPPAAL), and DSML, and highlights the challenges of achieving ex- act round-trip consistency. To ensure clarity, we presented concrete examples where one-to-one mappings were not feasible, such as the handling of extern variables, type precision mismatches (uint8 t and uint32 t), define directives, and enumerations, along with the strategies adopted to approximate or preserve these constructs. Further- more, Table 8 (Round-Trip Transformation Analysis) systematically documents these challenges, the mitigation approaches, and whether information loss occurred, thereby providing quantitative evidence of fidelity across transformations. Complementing this, Figure 5 (Round Trip Transformations) illustrates the notion of delta loss and demonstrates how information discrepancies may accumulate during round- trip conversions. To strengthen the manuscript, we have now high- lighted these discussions more prominently in the revised version so that the treatment of semantic equivalence, approximation strategies,

and information loss is immediately evident to the reader. Additionally, a deeper discussion of semantic equivalence issues and approximation strategies is essential to strengthen the framework. To address this, we have expanded the manuscript by adding a comprehensive discussion of semantic evaluation issues in the Discussion section. This new paragraph explicitly details how timing constructs in SystemVerilog are mapped to UPPAAL clocks, how preprocessor directives and data types are approximated across notations, and where information loss may occur in reverse transformations. We also explain the strategies adopted, such as preserving constructs as comments, type generalization, and programmatic recovery, to minimize delta loss and maintain semantic fidelity. This addition directly addresses the re- viewer’s concern by clarifying cases where exact semantic equivalence cannot be achieved, while outlining how our framework mitigates such discrepancies to ensure practical round-trip transformations.

R1.5 Case studies are appropriate, but evaluation metrics are too narrow. In addition to latency, memory, and transformation accuracy, please include productivity gains (developer time saved), defects detected or prevented due to synchronization, and scalability results (lines of code or model size handled). Comparisons with baseline workflows or existing frameworks would highlight practical improvements.

Author’s response: We sincerely thank the reviewer for this constructive suggestion. We fully agree that, beyond latency, memory, and transformation accuracy, broader evaluation metrics such as productivity gains, defect detection, and scalability are essential for demonstrating the practical impact of our framework. At this stage, the framework is still in its growing phase, we have now explicitly ad- dressed this point in the revised manuscript. A new paragraph has been added to the Future Work section (Section 7.2), where we outline our plan to conduct industrial-scale validation with four independent development teams. These studies will assess productivity improvements, synchronization-driven defect prevention, and scalability across large codebases. We believe this addition acknowledges the reviewer’s concern and sets a clear trajectory for the next phase of our research.

R1.6 Tool support and reproducibility need more clarity. Specify whether the tool supports round-trip editing in all notations, what dependencies (ANTLR version, UPPAAL integration) are required, and provide screenshots or GUI demonstration figures.

Author’s response: Your point is right. We have included this aspect in the Limitations section. At the current stage, our frame- work cannot perform direct external tool collaboration with UPPAAL, C compilers, or Simulink. For this purpose, users need to manually import the generated code into the respective working environments/tools. To clarify compatibility, we have now explicitly mentioned the supported versions: UPPAAL 4.1.26, GCC 11.2 (C compiler), SystemVerilog IEEE 1800-2017, and ANTLR 4.13.0. (Section 7.3). Additionally, GUI demonstration figures have already been pro- vided in Section 4 (Implementation Architecture). We cannot add more figures, since doing so would make the manuscript unnecessarily long and more like a user manual, a concern also raised in your sub- sequent comment regarding paper length. Instead, for reproducibility and further investigation, we have made all resources available in our GitHub repository (Ref 33), including the Eclipse RCP tool for direct usage and exploration.

R1.7 The manuscript is overly long. Detailed grammar rules and AST breakdowns could be moved to supplementary material, while keeping only key illustrative examples in the main text. Figures should have clearer captions explaining their relevance to the framework.

Author’s response: We sincerely thank the reviewer for this valuable observation. We acknowledge that the manuscript was lengthy in its earlier version, with detailed grammar rules and complete AST breakdowns contributing to excessive technical depth in the main text. In response, we have retained only key illustrative examples while re- locating full grammar specifications and AST expansions to Appendix A to enhance focus and clarity. Furthermore, we have excluded non- essential transformation rules from Section (Proof of Concept – Validation and Evaluation). Only the most representative and impactful rules are retained to emphasize conceptual clarity, maintain scientific precision, and enhance the overall coherence and conciseness of the pa- per. In addition to these structural refinements, the entire manuscript has been systematically revised to reduce content. Several related paragraphs have been merged into concise, cohesive discussions to improve narrative flow and scientific readability. The reduced version, with these refinements highlighted in the manuscript, now offers a more streamlined presentation without compromising technical completeness or scholarly rigor.

Furthermore, figure captions have been thoroughly revised and expanded to clearly convey their significance and linkage to the proposed framework.

R1.8 Limitations are understated. Explicitly discuss current lack of concurrency modeling, scalability to larger industrial systems, and potential challenges in heterogeneous hardware–software co-design. A short future work section would better position the research.

Author’s response: We thank the reviewer for this valuable observation. In the revised manuscript, we have expanded the Limitations section 7.1 to explicitly acknowledge the current lack of concurrency modeling (threads, tasks, interrupts), the challenges of scaling the framework to very large industrial systems, and the complexities of heterogeneous hardware–software co-design. These aspects are identified as critical but currently unsupported features of our framework. Furthermore, to better position this research, we have introduced a Future Work subsection 7.2, where we outline planned extensions including (i) incorporating concurrency constructs to enhance applicability for safety-critical domains, (ii) industrial-scale validation on larger case studies to systematically assess scalability, and (iii) exploring hybrid strategies to address cross-domain hardware–software co-design challenges. These additions provide a clearer picture of the framework’s present boundaries and its evolution roadmap.

R1.9 Overall, the paper has strong potential, but revisions are needed to clarify novelty, strengthen validation, and sharpen the technical depth of transformation and semantic fidelity claims.

Author’s response: Overall, we sincerely appreciate the reviewer’s detailed and constructive feedback. We acknowledge that while the paper has strong potential, revisions were indeed necessary to more clearly highlight the novelty of our framework, strengthen the validation with empirical evidence and case studies, and sharpen the discussion on transformation rules and semantic fidelity. We have accordingly revised the manuscript to address these points by (i) clarifying the uniqueness of our blended modeling approach, (ii) expanding validation with empirical coverage, scalability considerations, and round- trip analysis, and (iii) deepening the discussion of semantic equivalence issues and limitations. These revisions significantly enhance the technical depth, reproducibility, and positionin

---

## [Decision Letter · Decision Letter 1]

11 Nov 2025

A blended modeling framework for real-time design and verification of safety-critical embedded systems

PONE-D-25-39470R1

Dear Dr. Mehboob Awan,

We’re pleased to inform you that your manuscript has been judged scientifically suitable for publication and will be formally accepted for publication once it meets all outstanding technical requirements.

Kind regards,

Asadullah Shaikh, Ph.D.

Academic Editor

PLOS ONE

Additional Editor Comments (optional):

Reviewers' comments:

Reviewer's Responses to Questions

**Comments to the Author**

1. If the authors have adequately addressed your comments raised in a previous round of review and you feel that this manuscript is now acceptable for publication, you may indicate that here to bypass the “Comments to the Author” section, enter your conflict of interest statement in the “Confidential to Editor” section, and submit your "Accept" recommendation.

Reviewer #1: All comments have been addressed

Reviewer #4: All comments have been addressed

2. Is the manuscript technically sound, and do the data support the conclusions?

Reviewer #1: Yes

Reviewer #4: Yes

3. Has the statistical analysis been performed appropriately and rigorously?

Reviewer #1: Yes

Reviewer #4: Yes

4. Have the authors made all data underlying the findings in their manuscript fully available?

Reviewer #1: Yes

Reviewer #4: Yes

5. Is the manuscript presented in an intelligible fashion and written in standard English?

Reviewer #1: Yes

Reviewer #4: Yes

6. Review Comments to the Author

Reviewer #1: The authors have thoroughly addressed all the queries and concerns raised during the review process. With the necessary revisions made, the manuscript is now in its final form and can be accepted for publication.

Reviewer #4: (No Response)

7. PLOS authors have the option to publish the peer review history of their article (what does this mean?). If published, this will include your full peer review and any attached files.

Reviewer #1: No

Reviewer #4: No

---

## [Editor Report · Acceptance letter]

PONE-D-25-39470R1

PLOS ONE

Dear Dr. Mehboob Awan,

I'm pleased to inform you that your manuscript has been deemed suitable for publication in PLOS ONE. Congratulations! Your manuscript is now being handed over to our production team.

Kind regards,

on behalf of

Prof. Asadullah Shaikh

Academic Editor

PLOS ONE